# IS TRAINING NECESSARY FOR REPRESENTATION LEARNING

## ABSTRACT

The field of neural network-based encoders is currently experiencing rapid growth. However, in the pursuit of higher performance, models are becoming increasingly complex and specialized for specific datasets and tasks, resulting in a loss of generality. In response to this trend, we explore the finite element method (FEM) as a general solution for feature extraction and introduce LagrangeEmbedding, an untrainable encoder with a universal architecture across various types of raw data and recognition tasks. Our experimental results demonstrate its successful application and good performance in diverse domains, including data fitting, computer vision, and natural language processing. LagrangeEmbedding is explainable and adhering to the error-bound formula in FEM, which governs the relationship between mean absolute error (MAE) and the number of model parameters. As the encoder has no trainable parameters, neural networks utilizing it only need to train a linear layer. This reduces gradient computation and significantly accelerates training convergence. Our research promises to advance machine learning by opening up new avenues for unsupervised representation learning. Source code: `https://anonymous.4open.science/r/LagrangeEmbedding-652D`.

## 1 INTRODUCTION

In contrast to traditional unsupervised methods (Pearson, 1901; Sparck Jones, 1972), neural network-based encoders have shown their ability to extract useful features more effectively from raw data through deeper structures and a larger number of parameters. However, research in transfer learning has revealed a thought-provoking phenomenon: despite being trained for specific recognition tasks, these encoders exhibit unsupervised learning characteristics. In other words, while they are initially learned for a particular task, their feature extraction capabilities can be transferred to different recognition tasks, independent of labeled information. This raises a bold question: can unsupervised encoders potentially match the performance of neural network-based encoders?

Our research aims to construct an unsupervised encoder, which we named LagrangeEmbedding, which is capable of achieving performance comparable to neural network-based encoders, but with a universal architecture across different types of raw data and recognition tasks. Such an encoder holds important significance in the field of representation learning. The encoder obviates the need for training and fine-tuning when applied to various recognition tasks. Consequently, it offers lower model-training consumption and higher generality than neural network-based encoders.

To address this challenge, we opted for the finite element method as our theoretical guidance and employed the Lagrange basis from FEM as our encoder. This selection is grounded in two primary reasons. 1) The Lagrange basis is a one-to-one mapping, ensuring no information loss in feature extraction; 2) the Lagrange basis aims to depict the distribution of the raw data and is independent of the objective function to be fitted, therefore it is unsupervised. Any LagrangeEmbedding-based network can be written as a linear combination of Lagrange basis functions, which means these models only contain one linear layer with no bias and activation functions. As a result, when the loss function is convex, the empirical risk with respect to model parameters is also convex. Through a series of experiments, we show the advantage of convex risk that models typically complete training within 1 to 2 epochs. We also demonstrate that such an encoder can be successfully applied to various recognition domains, such as regression, image and text classification, and image super-resolution tasks.

Our research holds significant importance in the domain of unsupervised representation learning. Firstly, as a genuinely explainable and non-black box-like encoder, the structure of LagrangeEmbedding is deduced from a mature mathematical theory rather than determined through training experiments. Secondly, as an unsupervised encoder, LagrangeEmbedding exhibits extensive applicability across various recognition tasks. Lastly, the LagrangeEmbedding-based neural networks exhibit rapid training speeds. In summary, our research introduces novel, interpretable methods for unsupervised representation learning, simultaneously enhancing the generality and efficiency of neural networks.

## 2 LAGRANGEEMBEDDING

In the context of FEM, elements often serve as the fundamental building blocks of the triangulation *mesh*, taking the form of *simplices* created by connecting *nodes*. For instance, in 2D FEM, triangles with three nodes are commonly used, while 3D FEM often employs tetrahedra with four nodes. This concept is visually depicted in Figure 1 (left), where the mesh consists of eight nodes and seven triangles. This type of mesh is established by specifying the coordinates of discrete nodes and the vertex indices of simplices. Let $d$ represent the dimensions, $\{\boldsymbol{p}^{(i)}\}_{i=0}^{n-1}$ denote the grid nodes, and introduce a matrix $\boldsymbol{P}$ to store the node coordinates:

$$\boldsymbol{P}_{i,j} = \boldsymbol{p}_j^{(i)}.$$

Additionally, utilize a matrix $\boldsymbol{T}$ to store the indices of nodes constituting the simplices within the triangulation. Specifically, access the $j$-th sorted vertex of the $i$-th simplex in this mesh as $\boldsymbol{P}_{\boldsymbol{T}_{i,j},:}$. Figure 1 (left) illustrate a matrix $\boldsymbol{T}$ takes the following form:

$$\boldsymbol{T} = \begin{bmatrix} 6 & 0 & 3 & 2 & 7 & 3 & 4 \\ 7 & 6 & 7 & 3 & 4 & 4 & 3 \\ 5 & 5 & 1 & 1 & 5 & 7 & 2 \end{bmatrix}^T.$$

This matrix serves to describe all seven simplices within the mesh, such as the first simplex $\triangle\boldsymbol{p}^{(6)}\boldsymbol{p}^{(7)}\boldsymbol{p}^{(5)}$ and the last simplex $\triangle\boldsymbol{p}^{(4)}\boldsymbol{p}^{(3)}\boldsymbol{p}^{(2)}$.

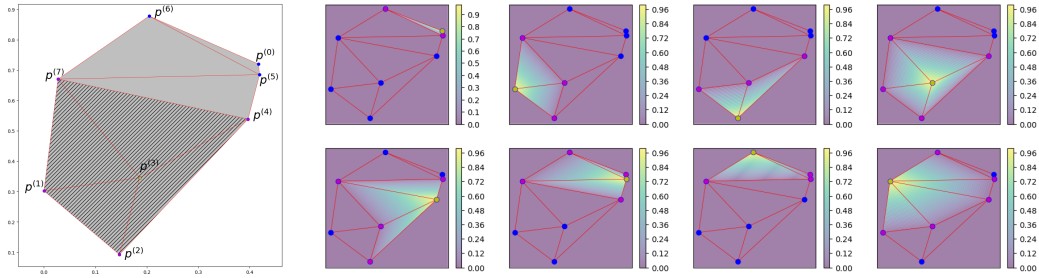

Figure 1: Left - Mesh with eight nodes and seven triangles. Right - Contours of eight Lagrange basis functions, linear variation of $\mathcal{L}_i$ associated with node $\boldsymbol{p}^{(i)}$ across all triangles.

The first-order Lagrange basis studied in this article, denoted as $\{\mathcal{L}_0(\boldsymbol{x}), \cdots, \mathcal{L}_{n-1}(\boldsymbol{x})\}$, are piecewise linear polynomials associated with nodes $\{\boldsymbol{p}^{(0)}, \cdots, \boldsymbol{p}^{(n-1)}\}$. These functions are defined such that $\mathcal{L}_i(\boldsymbol{p}^{(j)}) = \boldsymbol{1}_{i=j}$. Figure 1 (right) illustrates this: $\mathcal{L}_i(\boldsymbol{x})$ corresponds to node $\boldsymbol{p}^{(i)}$, exhibiting linear variation across all elements. Its support encompasses the union of all neighboring elements of node $\boldsymbol{p}^{(i)}$ (refer to the Appendix B for a 3-dimensional visualization). For example, $\text{supp}(\mathcal{L}_3) = \triangle\boldsymbol{p}^{(3)}\boldsymbol{p}^{(4)}\boldsymbol{p}^{(7)} \cup \triangle\boldsymbol{p}^{(3)}\boldsymbol{p}^{(7)}\boldsymbol{p}^{(1)} \cup \triangle\boldsymbol{p}^{(3)}\boldsymbol{p}^{(1)}\boldsymbol{p}^{(2)} \cup \triangle\boldsymbol{p}^{(3)}\boldsymbol{p}^{(2)}\boldsymbol{p}^{(4)}$.

In the realm of representation learning, Lagrange basis functions possess two highly valuable properties: the ability to simulate arbitrary neural networks and the capacity to calculate the similarity between raw data.

**Universal Approximation:** Each Lagrange basis function is globally continuous and piecewise linear, making the linear combination $f(\boldsymbol{x}; \boldsymbol{\theta}) = \sum_{i=0}^{n-1} \boldsymbol{\theta}_i \mathcal{L}_i(\boldsymbol{x})$ capable of approximating all continuous functions. In essence, this means that as long as a neural network can compute gradients

through backpropagation, it can be approximated with arbitrary precision by $f(\boldsymbol{x}; \boldsymbol{\theta})$. Given a function or pre-trained neural network $F(\boldsymbol{x})$, we can bound the approximation error as follows:

$$|f(\boldsymbol{x}; \boldsymbol{\theta}) - F(\boldsymbol{x})| \leq \max_{\boldsymbol{\xi} \in \Omega} \|\nabla f(\boldsymbol{\xi})\| \cdot h. \tag{1}$$

Here, $h = \max_i \max_j \mathbf{1}_{\text{dist}(\boldsymbol{p}^{(i)}, \boldsymbol{p}^{(j)})=1} \cdot \|\boldsymbol{p}^{(i)} - \boldsymbol{p}^{(j)}\|$ represents the maximum length of mesh edges and $\Omega$ is the domain over which the approximation occurs.

**Similarity Calculation:** Each Lagrange basis function is *Lipschitz continuous* due to its piecewise linear nature. This property enables the use of these functions for similarity calculations. For any given inputs $\boldsymbol{x}'$ and $\boldsymbol{x}''$, we have:

$$|\mathcal{L}_i(\boldsymbol{x}') - \mathcal{L}_i(\boldsymbol{x}'')| \leq \frac{1}{h_i} \|\boldsymbol{x}' - \boldsymbol{x}''\|.$$

In this equation, $h_i = \min_{\boldsymbol{p}^{(j)} \in \{\boldsymbol{p} | \mathbf{1}_{\text{dist}(\boldsymbol{p}^{(i)}, \boldsymbol{p})=1}\}} \|\boldsymbol{p}^{(i)} - \boldsymbol{p}^{(j)}\|$ represents the minimum length of edges that connecting to $\boldsymbol{p}^{(i)}$.

These two fundamental properties of Lagrange basis functions underscore their significance in representation learning. They enable the versatile approximation of complex functions and provide a structured approach to quantifying data similarity, guiding us in constructing our *Multiscale Domain Decomposition* method in the following subsection.

## 2.1 MULTISCALE DOMAIN DECOMPOSITION METHOD

In our earlier discussion, we introduced the Lagrange interpolation $f(\boldsymbol{x}; \boldsymbol{\theta})$ and its associated error-bound formula. However, this tool is not appropriate for machine learning modeling, since we face a crucial challenge: the "given function $F(\boldsymbol{x})$ to be fitted" represents a ground truth that remains unknown. Instead, in the scenario of machine learning, a typical dataset provides us with a collection of input-target pairs. For any given simplex, select a subset $\{(x^{(k_0)}, y^{(k_0)}), \cdots, (x^{(k_{m'-1})}, y^{(k_{m'-1})})\}$ from the training set $\{(x^{(0)}, y^{(0)}), \cdots, (x^{(m-1)}, y^{(m-1)})\}$ where $m'$ is the cardinality of subset, $m$ is the cardinality of subset, $\{k_i\}_{i=0}^{m'-1} \subseteq \{i\}_{i=0}^{m-1}$, and all subset elements reside within the given simplex. Our goal now is to assess the error of $f(\boldsymbol{x}; \boldsymbol{\theta})$ within this simplex.

---

**Algorithm 1** Domain decomposition method of generating multiscale mesh.

---

**Input:** Maximum degrees of freedom $N$ to perform. Depending on the size of the training set.
**Input:** Initial Simplex Indices Matrix $\boldsymbol{T}$ of shape $(1, d+1)$ with $\boldsymbol{T}_{0,i} = i$.
**Input:** Initial Node Matrix $\boldsymbol{P}$ containing coordinates of $d+1$ points forming a simplex covering all training raw data.
**Output:** The updated Node Matrix $\boldsymbol{P}$ and Simplex Indices Matrix $\boldsymbol{T}$ of the refined mesh.

- $n \leftarrow d + 1$
**while** $n < N$ **do**
  - Create the binary Longest Edge Matrix $\boldsymbol{M}$ where $\boldsymbol{M}_{i,j} = 1$ indicates that the $i$-th edge is the longest side of the $j$-th simplex.
  - Formulate the binary Edge Membership Matrix $\boldsymbol{E}$ where $\boldsymbol{E}_{i,j} = 1$ indicates that the $i$-th edge is a side of the $j$-th simplex.
  - Establish the binary Data-Simplex Membership Matrix $\boldsymbol{B}$ where $\boldsymbol{B}_{i,j} = 1$ signifies that the $i$-th raw data falls within the $j$-th simplex.
  - Compute the index of the priority edge:

$$\arg\min_i \frac{\sum_j \sum_k \boldsymbol{M}_{i,j} \boldsymbol{B}_{k,j}}{\max(\sum_j \sum_k \boldsymbol{E}_{i,j} \boldsymbol{B}_{k,j}, 1)}.$$

The priority edge is the longest side among many simplices, and these relevant simplices cover a substantial portion of the raw data.
  - Insert a new node at the midpoint of the priority edge and update the Node Matrix $\boldsymbol{P}$.
  - Update the Simplex Indices Matrix $\boldsymbol{T}$ and utilize it to update mesh edges.
  - $n \leftarrow n + 1$
**end while**

---

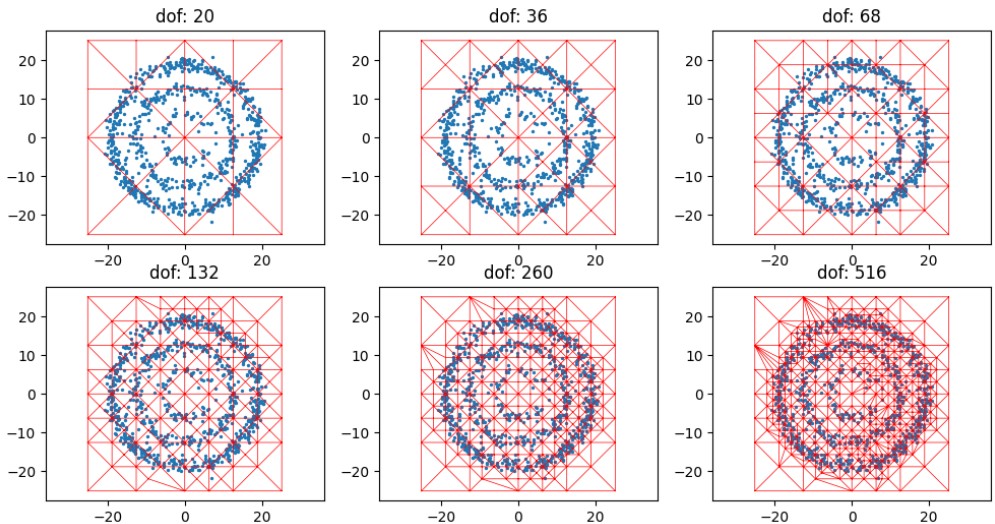

Figure 2: The images above depict a mesh variety during the iterations of Algorithm 1. Each triangle represents a 2-dimensional simplex, with increasing degrees of freedom indicating higher levels of refinement. Discrete points denote the raw data.

Crucially, due to the linearity of basis functions $\{\mathcal{L}_0(\boldsymbol{x}), \cdots, \mathcal{L}_{n-1}(\boldsymbol{x})\}$ within each simplex, their linear combination $f(\boldsymbol{x}; \boldsymbol{\theta})$ also remains linear within these simplices. As a result, in the given simplex, there exists a set of coefficients $\boldsymbol{\beta}$ that we can express $f(\boldsymbol{x}^{(i)}; \boldsymbol{\theta})$ as:

$$f(\boldsymbol{x}^{(i)}; \boldsymbol{\theta}) = \boldsymbol{\beta}_0 \boldsymbol{x}_0^{(i)} + \cdots + \boldsymbol{\beta}_{d-1} \boldsymbol{x}_{d-1}^{(i)} + \boldsymbol{\beta}_d.$$

Therefore we can obtain the following error bound by solving a *Ordinary Least Squares* problem

$$\sum_{i=0}^{m'-1} |y^{(i)} - f(\boldsymbol{x}^{(i)}; \boldsymbol{\theta})|^2 \le \sum_{i=0}^{m'-1} |y^{(i)} - (\hat{\boldsymbol{\beta}}_0 \boldsymbol{x}_0^{(i)} + \cdots + \hat{\boldsymbol{\beta}}_{d-1} \boldsymbol{x}_{d-1}^{(i)} + \hat{\boldsymbol{\beta}}_d)|^2, \tag{2}$$

where $\hat{\boldsymbol{\beta}} = (\boldsymbol{X}^T \boldsymbol{X})^{-1} \boldsymbol{X}^T \boldsymbol{y}$, $\boldsymbol{X}_{i,:d} = \boldsymbol{x}^{(i)}$, $\boldsymbol{X}_{i,d} = 1$, and $\boldsymbol{y}_i = y^{(i)}$. This formula shows the error bound reduced to 0 when $m' \le d + 1$. By combining this conclusion with the global error-bound formula Eqn. (1), we can summarize two critical goals for mesh generation in modeling:

1. Each simplex in the mesh should ideally contain as few original data points from the training set as possible. When each simplex covers no more than $d + 1$ raw data examples, the model perfectly fits the training set.

2. Decreasing the bound of mesh edge lengths results in a reduced bound of error.

Algorithm 1 is designed to achieve these two goals. It describes the process of generating a multiscale mesh and serves as the initial step of constructing the LagrangeEmbedding architecture. To enhance readability, this brief pseudocode traverses all training raw data, simplices, and edges in each iteration. For acceleration, we use divide-and-conquer techniques in the program.

In each iteration of Algorithm 1, a new fine node is added to the grid, and several coarse simplices are subdivided into more fine simplices. Figure 2 illustrates the process of refining a mesh. As the number (degrees of freedom) of nodes increases, each simplex (triangle) covers a reduced amount of raw data. Additionally, the side objective is to split the longest side of each simplex, progressively generating more acute triangles to minimize the value of $h$ in Eqn. (1).

## 2.2 INFERENCE VIA LAGRANGEEMBEDDING

In the preceding section, we introduced the multiscale domain decomposition method, a tool for initializing LagrangeEmbedding. Now, we formulate the Lagrange basis from its original definition

to establish the foundational architecture of LagrangeEmbedding. It is important to highlight that the traditional Lagrange basis involves unbalanced computing of barycentric coordinates, which may not be well-suited for parallel deep learning platforms (Appendix C details traditional definition of Lagrange basis). Consequently, in this subsection, we undertake a re-derivation of the Lagrange basis to better parallel computing.

Let $n_t$ represent the number of simplices in the multiscale mesh. We introduce the Parameters Tensor $\mathbf{S}$ defined as:

$$
\mathbf{S}_{j,:,:} = \begin{bmatrix} \boldsymbol{p}_0^{(\boldsymbol{T}_{j,0})} & \cdots & \boldsymbol{p}_{d-1}^{(\boldsymbol{T}_{j,0})} & 1 \\ \vdots & \ddots & \vdots & \vdots \\ \boldsymbol{p}_0^{(\boldsymbol{T}_{j,d-1})} & \cdots & \boldsymbol{p}_{d-1}^{(\boldsymbol{T}_{j,d-1})} & 1 \\ \boldsymbol{p}_0^{(\boldsymbol{T}_{j,d})} & \cdots & \boldsymbol{p}_{d-1}^{(\boldsymbol{T}_{j,d})} & 1 \end{bmatrix}^{-1}, \quad j = 0, \cdots, n_t - 1.
$$

Additionally, we introduce the Node Membership tensor $\mathbf{M}$ defined as:

$$
\mathbf{M}_{i,j,k} = \begin{cases} 1, & \text{if the } i\text{-th node matches the } k\text{-th vertex of the } j\text{-th simplex,} \\ 0, & \text{other cases.} \end{cases}
$$

By defining:

$$
\boldsymbol{U}_{j,k}(\boldsymbol{x}) = \sum_{\tau=0}^{d-1} \mathbf{S}_{j,\tau,k} \cdot \boldsymbol{x}_\tau + \mathbf{S}_{j,d,k}, \quad j = 0, \cdots, n_t - 1,\ k = 0, \cdots, d.
$$

We will demonstrate in Appendix A that the following function qualifies the definition of Lagrange basis:

$$
\mathcal{L}_i(\boldsymbol{x}) = \frac{\sum_{j=0}^{n_t-1} \sum_{k=0}^{d} \mathbf{1}_{\min_\tau \boldsymbol{U}_{j,\tau}(\boldsymbol{x}) \geq 0} \cdot \mathbf{M}_{i,j,k} \cdot \boldsymbol{U}_{j,k}(\boldsymbol{x})}{\max(\sum_{j=0}^{n_t-1} \sum_{k=0}^{d} \mathbf{1}_{\min_\tau \boldsymbol{U}_{j,\tau}(\boldsymbol{x}) \geq 0} \cdot \mathbf{M}_{i,j,k}, 1)}, \quad i = 0, \cdots, n - 1. \tag{3}
$$

So far we have successfully constructed LagrangeEmbedding:

$$
\text{Encoder} : \mathbb{R}^d \to [0, 1]^n,
$$
$$
\boldsymbol{x} \mapsto (\mathcal{L}_0(\boldsymbol{x}), \cdots, \mathcal{L}_{n-1}(\boldsymbol{x})).
$$

Finally, LagrangeEmbedding is original; in Appendix F we discuss the underlying principles that differ between it and kernel methods(Lee et al., 2017; Matthews et al., 2018; Kapoor et al., 2021).

## 3 EXPERIMENTS

In the upcoming sections, we present a comprehensive series of experiments to showcase the effectiveness and universality of LagrangeEmbedding across various tasks. Our exploration begins with an analysis of its performance in regression tasks, followed by examinations in image recognition and text recognition. Additionally, Appendix D introduces more details and other applications.

### 3.1 REGRESSION TASKS

#### 3.1.1 THE LIMITATIONS OF TRADITIONAL REGRESSORS AND NEURAL NETWORKS

Traditional regressors often struggle to overcome overfitting automatically, while neural networks have excelled in solving this problem but encounter difficulties in fitting multi-frequency functions, a challenge known as the *frequency principle* (Xu et al., 2019). Figure 3 vividly illustrates this conflict: on the left, SVR (Platt et al., 1999) successfully fits dataset $\mathbb{B}$ but overfits dataset $\mathbb{A}$, while the neural network fitting is the reverse, performing well on $\mathbb{A}$ but underfitting $\mathbb{B}$. This discrepancy shows the limitation of traditional regressors and neural networks in distinguishing whether a given raw data is in a high-frequency region or a high-noise region. As introduced in the preceding section, LagrangeEmbedding relies on a multiscale mesh, allowing it to adaptively fit data in high-frequency regions by employing fine simplices. Figure 3 (right) serves as a testament to the LagrangeEmbedding-based neural network's capability to learn the challenging dataset $\mathbb{C}$.

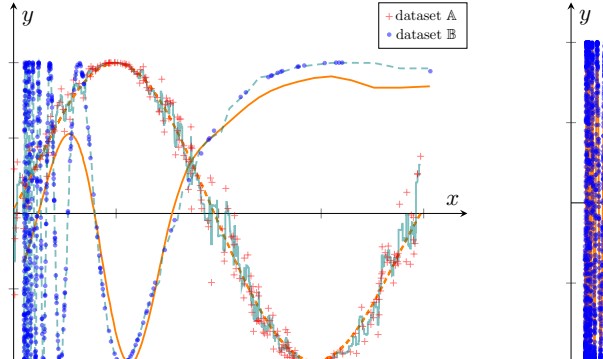 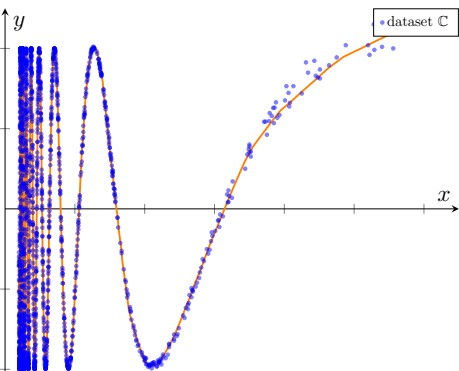

Figure 3: Left - illustrates the performance of traditional regressors in fitting the high-noise dataset $\mathbb{A} = \{(x,y)|X \sim u(0, \frac{\pi}{4}), Y \sim N(\sin 8x, |\cos 8x|)\}$ and the multi-frequency dataset $\mathbb{B} = \{(x,y)|\frac{1}{X} \sim u(0.02, 0.5), y = \sin \frac{1}{x}\}$. The dashed teal curve shows that the traditional regressor (*e.g.*, Support Vector Regression) succeeds in fitting dataset $\mathbb{B}$. However, when applied to the high-noise dataset $\mathbb{A}$, shown by the solid teal curve, the traditional regressor exhibits overfitting, struggling to generalize automatically. Conversely, we also showcase the performance of a neural network (*e.g.*, Multi-Layer Perceptron) in fitting dataset $\mathbb{A}$, exemplified by the dashed orange curve. However, as depicted by the solid orange curve, this neural network faces challenges in fitting multi-frequency dataset $\mathbb{B}$, indicative of underfitting. Right - We present the exceptional adaptability of a LagrangeEmbedding-based network in fitting the dataset $\mathbb{C} = \{(x,y)|\frac{1}{X} \sim u(0.02, 0.5), Y \sim N(\sin \frac{1}{x}, 0.5x^2)\}$, which is both high-noise and multi-frequency.

### 3.1.2 COMPARISON EXPERIMENTS

As mentioned earlier, neural networks often struggle to fit multi-frequency datasets effectively. Therefore, our primary focus is comparing the LagrangeEmbedding-based network with traditional regressors. To evaluate the effectiveness and generalization of the LagrangeEmbedding-based network, we have devised four diverse datasets, each generated from distinct probability distributions:

1. $\mathbb{A}^1$: Generated from the distribution $\{(x,y)|X \sim U(-\pi, \pi), Y \sim \mathcal{N}(\sin x, \frac{1}{5}\cos^2 x)\}$, with 1000 training examples and 200 test examples. The LagrangeEmbedding-based network was trained with a learning rate of 0.1.

2. $\mathbb{B}^1$: Generated from the distribution $\{(x,y)|\frac{1}{X} \sim U(0.02, 1.0), Y \sim \mathcal{N}(\sin \frac{1}{x}, 0.01)\}$, with 1000 training examples and 200 test examples. We trained the LagrangeEmbedding-based network with a learning rate of 0.9.

3. $\mathbb{A}^2$: Generated from the distribution $\{(\boldsymbol{x},y)|\boldsymbol{X}_i \sim U(-\pi, \pi), Y_i \sim \mathcal{N}(\sin \boldsymbol{x}_i, \frac{1}{10}\cos^2 \boldsymbol{x}_i)$ $,Y = \frac{1}{2}(Y_1 + Y_2)\}$, with 7,500 training examples and 1,500 test examples. The LagrangeEmbedding-based network was trained with a learning rate of 0.1.

4. $\mathbb{B}^2$: Generated from the distribution $\{(\boldsymbol{x},y)|\frac{1}{\boldsymbol{X}_i} \sim U(0.05, 0.5), Y_i \sim \mathcal{N}(\sin \frac{1}{\boldsymbol{x}_i}, 0.01), Y$ $= \frac{1}{2}(Y_1 + Y_2)\}$, with 50,000 training examples and 10,000 test examples. Training utilized a learning rate of 0.9.

Table 1 displays the coefficient of determination ($R^2$) scores for the LagrangeEmbedding-based network and traditional regressors (Thiel, 1950; Cantzler, 1981; Zhang, 2004; Hilt & Seegrist, 1977; Stone, 1974; Jain et al., 2018; Murphy, 2012; Platt et al., 1999; Friedman, 2001; Breiman, 2001) across fitting the four datasets. The LagrangeEmbedding-based network consistently achieves high $R^2$ scores across all test sets, demonstrating the effectiveness of the InterpolationNet on both high-noise and multi-frequency datasets. Furthermore, the minimal gap between training and test set evaluations underscores the robustness of the LagrangeEmbedding-based network, indicating its capability of generalization.

Table 1: A comprehensive comparison between the LagrangeEmbedding-based network and traditional regressors. The left half of each paired column displays the training $R^2$ score, while the right half showcases the corresponding test $R^2$ score.

| METHOD | $\mathbb{A}^1$ | | $\mathbb{B}^1$ | | $\mathbb{A}^2$ | | $\mathbb{B}^2$ | |
|---|---|---|---|---|---|---|---|---|
| OLS Linear | 0.037 | 0.042 | 0.963 | 0.951 | 0.085 | 0.092 | 0.984 | 0.984 |
| Theil-Sen | -44.7 | -54.4 | 0.958 | 0.946 | -0.41 | -3.81 | 0.982 | 0.982 |
| RANSAC | -1.21 | -1.43 | 0.963 | 0.951 | -27.0 | -27.1 | 0.983 | 0.983 |
| Huber | 0.036 | 0.041 | 0.962 | 0.949 | 0.085 | 0.092 | 0.984 | 0.984 |
| Ridge | 0.031 | 0.038 | 0.963 | 0.951 | 0.055 | 0.061 | 0.984 | 0.984 |
| RidgeCV | 0.037 | 0.042 | 0.963 | 0.951 | 0.085 | 0.092 | 0.984 | 0.984 |
| SGD | 0.009 | 0.01 | 0.962 | 0.95 | 0.005 | 0.004 | 0.983 | 0.983 |
| KRR | 0.0036 | 0.04 | 0.97 | 0.962 | 0.056 | 0.051 | 0.993 | 0.992 |
| SVR | 0.11 | 0.101 | 0.97 | 0.962 | 0.29 | 0.308 | 0.992 | 0.992 |
| Gradient Boosting | 0.964 | 0.962 | 0.98 | 0.96 | 0.989 | 0.988 | 0.992 | 0.99 |
| Random Forests | 1.0 | 0.999 | 0.995 | 0.945 | 1.0 | 0.999 | 0.999 | 0.99 |
| Voting | 0.852 | 0.852 | 0.942 | 0.917 | 0.869 | 0.868 | 0.951 | 0.946 |
| LagrangeEmbedding Net | 1.0 | 1.0 | 0.971 | 0.963 | 0.999 | 0.999 | 0.992 | 0.992 |

### 3.1.3 EXPERIMENTAL VALIDATION OF ERROR-BOUND FORMULA

Our LagrangeEmbedding exhibits the unique advantage of being explainable, distinguishing it from most network-based encoders with a black-box nature. Its performance is predictable even before learning. To illustrate this property, we conducted experiments of fitting two objective functions: $y = \sin\frac{1}{x}$ and $y = \sum_{i=1}^{2} \sin\frac{x_i}{2\pi}$. Subsequently, we evaluated the test $l^2$ errors. The results depicted in Figure 4 provide compelling evidence that $\frac{1}{m}\sum_{i=0}^{m-1}|f(\boldsymbol{x}^{(i)};\boldsymbol{\theta}) - y^{(i)}|^2 = O(n_t^{-1/d})$. Since $(n_t/d!)^{1/d} = O(h^{-1})$, we experimentally prove that the LagrangeEmbedding-based network $f(\boldsymbol{x};\boldsymbol{\theta})$ holds the error-bound formula Eqn. (1).

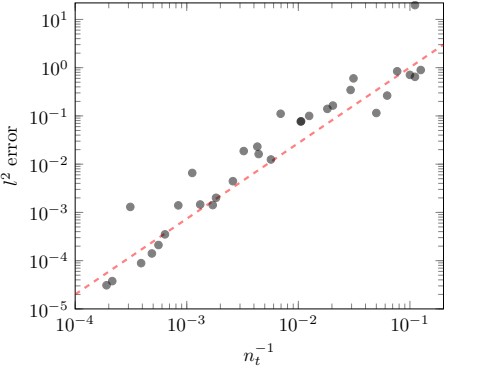 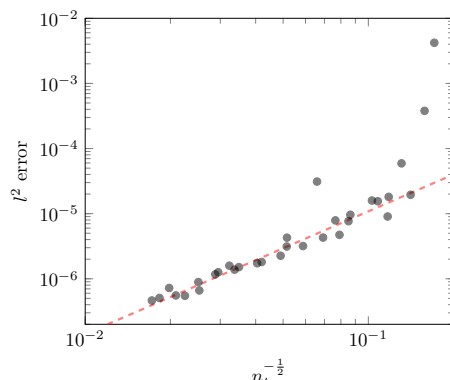

Figure 4: Left - the results of 32 experiments fitting a 1-dimensional function $y = \sin\frac{1}{x}$, with each gray point representing an experiment and showing the relationship between $n_t^{-1}$ and Mean Square Error (MSE). Right - 32 experiments fitting a 2-dimensional function $y = \sum_{i=1}^{2} \sin\frac{x_i}{2\pi}$, mirroring the left side with gray points signifying individual experiments and demonstrating the relationship between $n_t^{-1/2}$ and MSE.

## 3.2 COMPUTER VISION

In this section, our attention turns towards verifying the generalization of our encoder across different recognition tasks. Furthermore, we demonstrate the advantage of LagrangianEmbedding-based networks in faster training.

### 3.2.1 PREPROCESSING LAYER

In this section, we address the application of LagrangeEmbedding for feature extraction from images, using the basic MNIST dataset as an example. If we employ LagrangeEmbedding to extract features from images directly, it will map $\mathbb{R}^{784}$ to $[0,1]^n$, and the computational consumption will become extremely large. Since $Error = O(h) = O(n^{-1/784})$, each twofold improvement in performance or halved training MSE loss requires an exponential increase in the number of model parameters to $2^{784}$ times. In theory, as a universal encoder, LagrangeEmbedding could handle the most complex recognition tasks, for example, classifying images on the MNIST dataset with $2^{784}$ different classes. Therefore, for the real MNIST dataset, LagrangeEmbedding is a heavyweight encoder. To address the problem, we introduce an untrainable preprocessing layer that compresses the raw image $\boldsymbol{x}$ into a series of one-dimensional data $\boldsymbol{X}$:

$$\begin{cases} \mathbf{Y}_{i,j,k} = \sum_{p=0}^{2} \sum_{q=0}^{2} \mathbf{W}_{p,q,k} \boldsymbol{x}_{28(2i+p)+2j+q}, & i,j = 0, \cdots 12; k = 0,1,2,3, \\ \mathbf{Z}_{i,j,k,l} = \sum_{p=0}^{2} \sum_{q=0}^{2} \mathbf{W}_{p,q,l} \mathbf{Y}_{2i+p,2j+q,k}, & i,j = 0, \cdots 5; k,l = 0,1,2,3, \\ \boldsymbol{X}_{96i+16j+4k+l,0} = \mathbf{Z}_{i,j,k,l}, & i,j = 0, \cdots 5; k,l = 0,1,2,3, \end{cases}$$

where the binary tensor $\mathbf{W}$ has values: $\mathbf{W}_{:,1,0} = \mathbf{W}_{1,0,1} = \mathbf{W}_{i,i,2} = \mathbf{W}_{i,2-i,3} = 1$.

**Remark:** This preprocessing layer contains no trainable model parameters. Empirically, the model's performance does not significantly change when modifying $\mathbf{W}$. This preprocessing layer is not the only way of dimensionality reduction.

### 3.2.2 EXPERIMENTAL VALIDATION OF ENCODER GENERALIZATION

To demonstrate the generalization of LagrangeEmbedding for various image recognition tasks, we introduce the neural network architecture depicted in Figure 5. Both the preprocessing layer and LagrangeEmbedding have non-trainable model parameters. The branch for image classification employs a linear layer, while the branch for image super-resolution utilizes another linear layer.

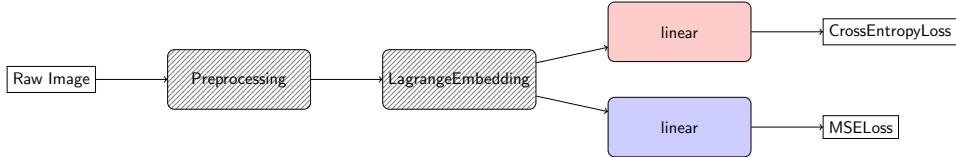

Figure 5: The network architecture comprises two branches: one dedicated to image classification and the other to image super-resolution. All shadowed blocks are frozen and untrainable.

We set the degree of freedom $n$ of LagrangeEmbedding to 16. For the linear layer dedicated to image classification, we employ the Adam optimizer to minimize cross-entropy loss with a learning rate of 0.001 and a batch size of 64 during training. After the first epoch, the model achieves a test accuracy exceeding 96%, reaching a peak accuracy of 97.25% within ten epochs. As for the linear layer used for image super-resolution, we utilize the Adam optimizer to minimize MSE loss with a learning rate of 0.0001 and a batch size of 64 for training. The target output is high-resolution images with size (32, 32), bicubic interpolated from raw images. After the first epoch, the model reaches a minimum test MSE of 0.0003, then no longer decline in the subsequent four epochs.

Post-testing, the LagrangeEmbedding-based network, which we used as an example, demonstrates performance roughly on par with a 6-layer CNN comprising 0.42 million parameters. The distinctive advantage of LagrangeEmbedding lies in its ability to limit neural network architecture to just one linear layer, typically completing training within one epoch, faster than other models.

### 3.3 NATURAL LANGUAGE PROCESSING

In this section, we delve into the practical application of LagrangeEmbedding for text feature extraction using the AG News dataset (Zhang et al., 2015) for classification tasks. Our experimental findings reveal that LagrangeEmbedding can directly extract features from raw text. However, recognizing that tokens are unordered categorical variables, to enhance performance, we add a preprocessing layer that converts each token to a four-dimensional vector, the proportion of the token appearing in four categories.

Throughout our experiment, we set the degree of freedom $n$ of LagrangeEmbedding to 64. We employ the SGD optimizer to minimize the cross-entropy loss, initiating with a learning rate of 5.0 and reducing it by a factor of 0.1 every two epochs. The batch size is set to 32. This experiment showcases the neural network achieving 90.01% test accuracy after the first epoch, with 90.4% test accuracy reached within just five epochs.

The LagrangeEmbedding-based network possesses a unique characteristic in text classification tasks: the number of its parameters remains independent of the token count. Compared to word2vec-based networks, our model performs equally well in classification but boasts only 256 model parameters, a significant reduction from the word2vec-based network with more than 6.13 million parameters.

## 4 FUTURE DIRECTIONS

It is essential to acknowledge that LagrangeEmbedding is not without limitations. Firstly, our multiscale domain decomposition method for initializing LagrangeEmbedding faces challenges in terms of parallelization and acceleration. In our implementation, this algorithm still runs in a non-parallel mode. Secondly, LagrangeEmbedding is a heavyweight tool for extracting features because it always considers the most complex scenarios. For instance, when loading a 24-bit RGB image with a size of $(1024, 1024)$, the LagrangeEmbedding doesn't know the underly number of classification categories, and it always tries to extract features for the classification task with $256^{1024^2}$ categories. Consequently, when doing an easy recognition task with high dimensional raw data, the LagrangeEmbedding will waste huge computing resources. As shown in section 3.2.1, our solution is adding an untrainable preprocessing layer to the model for reducing the dimension of raw data.

Future research should focus on developing efficient multiscale domain decomposition methods and better preprocessing layers.

## 5 CONCLUSION

We believe that when the performance of unsupervised encoders starts to approach or surpass that of neural network-based encoders, training will no longer be necessary for representation learning.

In an era marked by the rapid evolution of neural network-based encoders, our exploration began by recognizing their undeniable efficacy in extracting meaningful features from raw data. However, a thought-provoking revelation emerged from the study of transfer learning: these task-specific encoders exhibit unsupervised characteristics. This revelation prompted us to delve into the audacious question of whether unsupervised encoders could reach the performance of those neural network-based encoders.

Our response to this question culminated in the creation of LagrangeEmbedding, a revolutionary unsupervised encoder, which has a universal architecture capable of accommodating diverse raw data types and recognition tasks. LagrangeEmbedding extracts features by depicting the distribution of raw data, eliminating the need to consider the underlying meaning of raw data. This inherent characteristic grants it the property of generalization. As demonstrated in section 3.2.1, it efficiently extracts features from the output of a meaningless preprocessing layer.

Our experiments demonstrated the advantages of LagrangeEmbedding-based networks in several key aspects: 1) An unparalleled level of explainability. Our experimental results show that the performance of such models is in perfect agreement with the theoretical bound error formula, which demystifies the representation learning process. 2) Quick training. These models contain only one linear layer for training. They usually converged within a mere 1 to 2 epochs, challenging the conventional wisdom that the training method is essential for representation learning.

In conclusion, our research introduces a novel encoder derived from mature mathematical theory, rather than being proposed through extensive experimental trials. It stands as an explainable and non-black-box-like encoder, challenging long-standing assumptions about the role of training in feature extraction. By obviating the need for extensive training and fine-tuning, this encoder holds high value in representation learning research.

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

## A  PROOF OF LAGRANGE BASIS EXPRESSION

We will now demonstrate, in three concise steps, that Eqn. (3) qualifies as a Lagrange basis function.
**Piecewise Linear:** Since $\{U_{j,0}, \cdots, U_{j,d}\}$ are piecewise linear functions, their linear combination $L_i$ is also piecewise linear.
**Kronecker Delta:** From the definition of $U$ and $\mathsf{S}$, we have the following equation:

$$
[\boldsymbol{x}_0 \quad \cdots \quad \boldsymbol{x}_{d-1} \quad 1] = [\boldsymbol{U}_{j,0}(\boldsymbol{x}) \quad \cdots \quad \boldsymbol{U}_{j,d}(\boldsymbol{x})]
\begin{bmatrix}
\boldsymbol{p}_0^{(\boldsymbol{T}_{j,0})} & \cdots & \boldsymbol{p}_{d-1}^{(\boldsymbol{T}_{j,0})} & 1 \\
\vdots & \ddots & \vdots & \vdots \\
\boldsymbol{p}_0^{(\boldsymbol{T}_{j,d-1})} & \cdots & \boldsymbol{p}_{d-1}^{(\boldsymbol{T}_{j,d-1})} & 1 \\
\boldsymbol{p}_0^{(\boldsymbol{T}_{j,d})} & \cdots & \boldsymbol{p}_{d-1}^{(\boldsymbol{T}_{j,d})} & 1
\end{bmatrix}.
$$

Decomposing this equation, we obtain $\boldsymbol{x} = \sum_{k=0}^{d} \boldsymbol{U}_{j,k}(\boldsymbol{x})\boldsymbol{p}^{(\boldsymbol{T}_{j,k})}$ and $\sum_{k=0}^{d} \boldsymbol{U}_{j,k}(\boldsymbol{x}) = 1$. This implies two important conclusions: $\boldsymbol{U}_{j,k'}(\boldsymbol{p}^{(\boldsymbol{T}_{j,k''})}) = \mathbf{1}_{k'=k''}$ and $\min_\tau \boldsymbol{U}_{j,\tau}(\boldsymbol{x}) \geq 0$ is true if and only if $\boldsymbol{x}$ belongs to the $j$-th simplex. Therefore, we have

$$
\begin{cases}
\mathbf{1}_{\min_\tau \boldsymbol{U}_{j,\tau}(\boldsymbol{p}^{(i)})\geq 0} = \mathsf{M}_{i,j,k} = \boldsymbol{U}_{j,k}(\boldsymbol{p}^{(i)}) = 1, & \text{if } i = \boldsymbol{T}_{j,k} \\
\mathsf{M}_{i,j,k} = \boldsymbol{U}_{j,k}(\boldsymbol{p}^{(i)}) = 0, & \text{if } i \neq \boldsymbol{T}_{j,k}
\end{cases}
$$

This proves that $\mathcal{L}_i(\boldsymbol{p}^{(j)}) = \mathbf{1}_{i=j}$.
**Globally Continuity:** Lastly, since $\mathcal{L}_i$ is inherently linear within all simplices and exhibits continuity across all grid nodes, we can conclude that $\mathcal{L}_i$ is globally continuous.

## B  VISUALIZATION OF LAGRANGIAN BASIS

In section 2, we introduced first-order Lagrange basis functions, a set of piecewise linear functions defined on a mesh. Each basis function corresponds to a node.

Consider the grid depicted in Figure 6 (left). Taking the node $\boldsymbol{p}^{(20)}$ as an example, it has a total of four neighboring nodes: $\boldsymbol{p}^{(5)}$, $\boldsymbol{p}^{(0)}$, $\boldsymbol{p}^{(7)}$, and $\boldsymbol{p}^{(4)}$. By connecting these nodes, we can determine the support of the basis function $\mathcal{L}_{20}$.

In Figure 6 (middle), we present the function graphs of $\mathcal{L}_{20}$ and $\mathcal{L}_7$. It can be observed that these functions exhibit linear variations on each mesh triangle. Taking $\mathcal{L}_{20}$ as an example, its function value at $\boldsymbol{p}_{20}$ is 1, and 0 at all other nodes. Similarly, $\mathcal{L}_7$ has a function value of 1 at $\boldsymbol{p}_7$ and 0 at other nodes.

In Figure 6 (right), the orange triangles represent the function graph of $f(\boldsymbol{x}; \boldsymbol{\theta})$ on the domain $\triangle\boldsymbol{p}^{(0)}\boldsymbol{p}^{(7)}\boldsymbol{p}^{(9)}$, where the function values of $f(\boldsymbol{x}; \boldsymbol{\theta})$ at the vertices $(\boldsymbol{p}^{(0)}, \boldsymbol{p}^{(7)}, \boldsymbol{p}^{(9)})$ are $(\boldsymbol{\theta}_0, \boldsymbol{\theta}_7, \boldsymbol{\theta}_9)$, respectively. The green dots represent a subset of training set, where the projections (raw data) fall on $\triangle\boldsymbol{p}^{(0)}\boldsymbol{p}^{(7)}\boldsymbol{p}^{(9)}$. As shown in equation (2), when the number of green points does not exceed three, there exists a solution of $(\boldsymbol{\theta}_0, \boldsymbol{\theta}_7, \boldsymbol{\theta}_9)$ such that all green points lie on the surface of $f(\boldsymbol{x}; \boldsymbol{\theta})$, such that MSE reach a minimum value of 0.

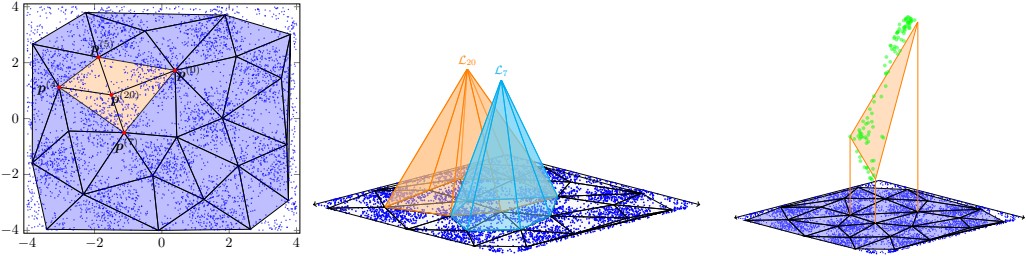

Figure 6: Data visualization. Left - an example of 2-dimensional mesh. Middle - the graphs of basis functions $\mathcal{L}_{20}$ and $\mathcal{L}_7$. Right - the graphs of the function $f(\boldsymbol{x}; \boldsymbol{\theta})$ and a subset of training set.

## C  THE TRADITIONAL EXPRESSION OF LAGRANGE BASIS

Given the complexity of FEM as a numerical method, to enhance understanding, we start with a two-dimensional case and a triangulated mesh to illustrate the traditional expression of basis functions. Let triangle $\triangle$ be defined by nodes $\{\boldsymbol{p}^{(i)}, \boldsymbol{p}^{(j)}, \boldsymbol{p}^{(k)}\}$. The following barycentric coordinates $\{\lambda_{\triangle,i}, \lambda_{\triangle,j}, \lambda_{\triangle,k}\}$ are three first-degree polynomials of $\boldsymbol{x}$

$$\begin{bmatrix} \lambda_{\triangle,i}(\boldsymbol{x}) \\ \lambda_{\triangle,j}(\boldsymbol{x}) \\ \lambda_{\triangle,k}(\boldsymbol{x}) \end{bmatrix} = \begin{bmatrix} \boldsymbol{p}_0^{(i)} & \boldsymbol{p}_0^{(j)} & \boldsymbol{p}_0^{(k)} \\ \boldsymbol{p}_1^{(i)} & \boldsymbol{p}_1^{(j)} & \boldsymbol{p}_1^{(k)} \\ 1 & 1 & 1 \end{bmatrix}^{-1} \begin{bmatrix} \boldsymbol{x}_0 \\ \boldsymbol{x}_1 \\ 1 \end{bmatrix}.$$

Referring to the instance depicted in Fig. 1 (left), the mesh consists of eight nodes and seven triangles. Specifically, let $\{\triangle^{(j)} = \triangle\boldsymbol{p}^{(T_{j,0})}\boldsymbol{p}^{(T_{j,1})}\boldsymbol{p}^{(T_{j,2})}|j = 0, \cdots, 6\}$. We will now verify that the following $\psi_3$ corresponds to the third basis function in this mesh

$$\psi_3(\boldsymbol{x}) = \frac{1}{\max(\sum_{i \in \{2,3,6,5\}} \mathbf{1}_{\boldsymbol{x} \in \triangle^{(i)}}, 1)} \sum_{i \in \{2,3,6,5\}} \mathbf{1}_{\boldsymbol{x} \in \triangle^{(i)}} \lambda_{\triangle^{(i)},3}(\boldsymbol{x}).$$

First, $\psi_3$ possesses values of *Kronecker Delta*:

$$\psi_3(\boldsymbol{x}) = \begin{cases} \frac{1}{\max(0,1)}(0 \cdot \lambda_{\triangle^{(2)},3}(\boldsymbol{x}) + 0 \cdot \lambda_{\triangle^{(3)},3}(\boldsymbol{x}) + 0 \cdot \lambda_{\triangle^{(6)},3}(\boldsymbol{x}) + 0 \cdot \lambda_{\triangle^{(5)},3}(\boldsymbol{x})) = 0, & \text{if } \boldsymbol{x} = \boldsymbol{p}^{(0)}, \\ \frac{1}{\max(2,1)}(1 \cdot 0 \quad\quad + 1 \cdot 0 \quad\quad + 0 \cdot \lambda_{\triangle^{(6)},3}(\boldsymbol{x}) + 0 \cdot \lambda_{\triangle^{(5)},3}(\boldsymbol{x})) = 0, & \text{if } \boldsymbol{x} = \boldsymbol{p}^{(1)}, \\ \frac{1}{\max(2,1)}(0 \cdot \lambda_{\triangle^{(2)},3}(\boldsymbol{x}) + 1 \cdot 0 \quad\quad + 1 \cdot 0 \quad\quad + 0 \cdot \lambda_{\triangle^{(5)},3}(\boldsymbol{x})) = 0, & \text{if } \boldsymbol{x} = \boldsymbol{p}^{(2)}, \\ \frac{1}{\max(4,0)}(1 \cdot 1 \quad\quad + 1 \cdot 1 \quad\quad + 1 \cdot 1 \quad\quad + 1 \cdot 1) \quad\quad = 1, & \text{if } \boldsymbol{x} = \boldsymbol{p}^{(3)}, \\ \frac{1}{\max(2,1)}(0 \cdot \lambda_{\triangle^{(2)},3}(\boldsymbol{x}) + 0 \cdot \lambda_{\triangle^{(3)},3}(\boldsymbol{x}) + 1 \cdot 0 \quad\quad + 1 \cdot 0) \quad\quad = 0, & \text{if } \boldsymbol{x} = \boldsymbol{p}^{(4)}, \\ \frac{1}{\max(0,1)}(0 \cdot \lambda_{\triangle^{(2)},3}(\boldsymbol{x}) + 0 \cdot \lambda_{\triangle^{(3)},3}(\boldsymbol{x}) + 0 \cdot \lambda_{\triangle^{(6)},3}(\boldsymbol{x}) + 0 \cdot \lambda_{\triangle^{(5)},3}(\boldsymbol{x})) = 0, & \text{if } \boldsymbol{x} = \boldsymbol{p}^{(5)}, \\ \frac{1}{\max(0,1)}(0 \cdot \lambda_{\triangle^{(2)},3}(\boldsymbol{x}) + 0 \cdot \lambda_{\triangle^{(3)},3}(\boldsymbol{x}) + 0 \cdot \lambda_{\triangle^{(6)},3}(\boldsymbol{x}) + 0 \cdot \lambda_{\triangle^{(5)},3}(\boldsymbol{x})) = 0, & \text{if } \boldsymbol{x} = \boldsymbol{p}^{(6)}, \\ \frac{1}{\max(2,1)}(1 \cdot 0 \quad\quad + 0 \cdot \lambda_{\triangle^{(3)},3}(\boldsymbol{x}) + 0 \cdot \lambda_{\triangle^{(6)},3}(\boldsymbol{x}) + 1 \cdot 0) \quad\quad = 0, & \text{if } \boldsymbol{x} = \boldsymbol{p}^{(7)}, \end{cases}$$

Then, $\psi_3$ is a first-degree polynomial in every triangle and supp $\psi_3 = \text{inn } \cup_{i \in \{2,3,6,5\}} \triangle^{(i)}$:

$$\psi_3(\boldsymbol{x}) = \begin{cases} \frac{1}{\max(0,1)}(0 \cdot \lambda_{\triangle^{(2)},3}(\boldsymbol{x}) + 0 \cdot \lambda_{\triangle^{(3)},3}(\boldsymbol{x}) + 0 \cdot \lambda_{\triangle^{(6)},3}(\boldsymbol{x}) + 0 \cdot \lambda_{\triangle^{(5)},3}(\boldsymbol{x})) = 0, & \text{if } \boldsymbol{x} \in \text{inn } \boldsymbol{T}_0, \\ \frac{1}{\max(0,1)}(0 \cdot \lambda_{\triangle^{(2)},3}(\boldsymbol{x}) + 0 \cdot \lambda_{\triangle^{(3)},3}(\boldsymbol{x}) + 0 \cdot \lambda_{\triangle^{(6)},3}(\boldsymbol{x}) + 0 \cdot \lambda_{\triangle^{(5)},3}(\boldsymbol{x})) = 0, & \text{if } \boldsymbol{x} \in \text{inn } \boldsymbol{T}_1, \\ \frac{1}{\max(1,1)}(1 \cdot \lambda_{\triangle^{(2)},3}(\boldsymbol{x}) + 0 \cdot \lambda_{\triangle^{(3)},3}(\boldsymbol{x}) + 0 \cdot \lambda_{\triangle^{(6)},3}(\boldsymbol{x}) + 0 \cdot \lambda_{\triangle^{(5)},3}(\boldsymbol{x})) = \lambda_{\triangle^{(2)},3}(\boldsymbol{x}), & \text{if } \boldsymbol{x} \in \text{inn } \boldsymbol{T}_2, \\ \frac{1}{\max(1,1)}(0 \cdot \lambda_{\triangle^{(2)},3}(\boldsymbol{x}) + 1 \cdot \lambda_{\triangle^{(3)},3}(\boldsymbol{x}) + 0 \cdot \lambda_{\triangle^{(6)},3}(\boldsymbol{x}) + 0 \cdot \lambda_{\triangle^{(5)},3}(\boldsymbol{x})) = \lambda_{\triangle^{(3)},3}(\boldsymbol{x}), & \text{if } \boldsymbol{x} \in \text{inn } \boldsymbol{T}_3, \\ \frac{1}{\max(0,1)}(0 \cdot \lambda_{\triangle^{(2)},3}(\boldsymbol{x}) + 0 \cdot \lambda_{\triangle^{(3)},3}(\boldsymbol{x}) + 0 \cdot \lambda_{\triangle^{(6)},3}(\boldsymbol{x}) + 0 \cdot \lambda_{\triangle^{(5)},3}(\boldsymbol{x})) = 0, & \text{if } \boldsymbol{x} \in \text{inn } \boldsymbol{T}_4, \\ \frac{1}{\max(1,1)}(0 \cdot \lambda_{\triangle^{(2)},3}(\boldsymbol{x}) + 0 \cdot \lambda_{\triangle^{(3)},3}(\boldsymbol{x}) + 0 \cdot \lambda_{\triangle^{(6)},3}(\boldsymbol{x}) + 1 \cdot \lambda_{\triangle^{(5)},3}(\boldsymbol{x})) = \lambda_{\triangle^{(5)},3}(\boldsymbol{x}), & \text{if } \boldsymbol{x} \in \text{inn } \boldsymbol{T}_5, \\ \frac{1}{\max(1,1)}(0 \cdot \lambda_{\triangle^{(2)},3}(\boldsymbol{x}) + 0 \cdot \lambda_{\triangle^{(3)},3}(\boldsymbol{x}) + 1 \cdot \lambda_{\triangle^{(6)},3}(\boldsymbol{x}) + 0 \cdot \lambda_{\triangle^{(5)},3}(\boldsymbol{x})) = \lambda_{\triangle^{(6)},3}(\boldsymbol{x}), & \text{if } \boldsymbol{x} \in \text{inn } \boldsymbol{T}_6, \end{cases}$$

Finally, since $\psi_3$ is continuous in all nodes and first-degree in all triangles, it is globally continuous.

# D  ADDITIONAL EXPERIMENTS AND APPLICATIONS

## D.1  FITTING HIGH-NOISE DATASET

In this section, we conduct the LagrangeEmbedding-based network on fitting the dataset $\mathbb{A} = \{(x, y)|X \sim U(-4, 4), Y \sim \mathcal{N}(\sin x, 0.2\cos^2 x)\}$. A comprises 6000 examples, with 5000 for training and 1000 for testing. Figure 7 shows the training progress.

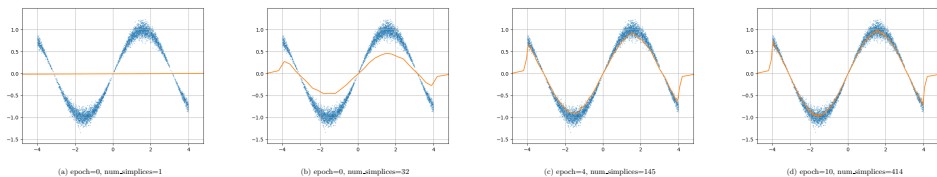

Figure 7: Blue dots represent the training set, while the orange curve represents the network.

## D.2  FITTING MULTI-FREQUENCY DATASET

In this section, we conduct the LagrangeEmbedding-based network on fitting the dataset $\mathbb{A} = \{(x, y)|\frac{1}{x} \sim U(0.02, 0.5), y = \sin \frac{1}{x}\}$. A comprises 6000 examples, with 5000 for training and 1000 for testing. Figure 8 illustrates the training progress. Remarkably, after just 4 epochs of training, the neural network outputs closely approximate the target values. By the 32nd epoch's conclusion, the neural network outputs and target values are nearly indistinguishable.

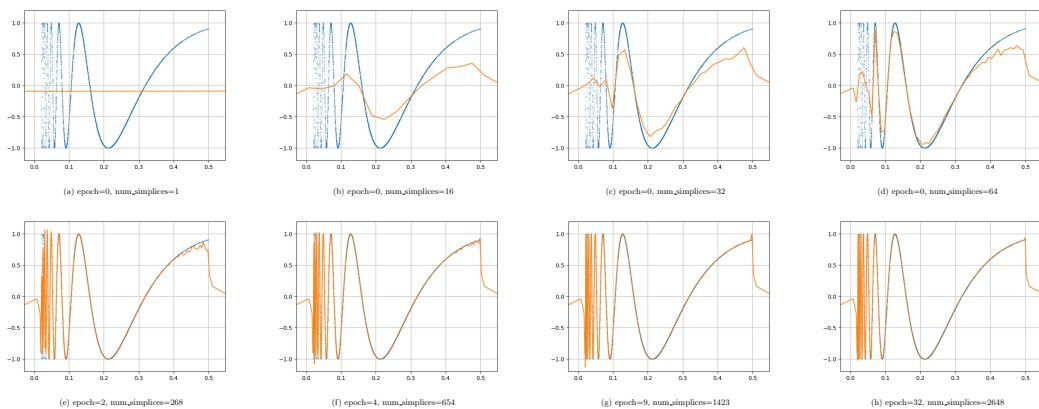

Figure 8: Blue dots represent the training set, while the orange curve represents the network.

## D.3  FIT A VECTOR-VALUED FUNCTION

In this instance, we utilize the LagrangeEmbedding-based network to fit spherical harmonics. Our dataset denoted as $\mathbb{A} = \{(\boldsymbol{x}, \boldsymbol{y})|\boldsymbol{x} = (\theta, \phi), \boldsymbol{y} = (\text{Real}(Y_4^2(\theta, \phi)), \text{Imag}(Y_4^2(\theta, \phi))), \Theta \sim U(0, 2\pi), \Phi \sim U(0, \pi)\}$, comprises 48,000 examples, with 40,000 allocated for training and an additional 8,000 for testing. Figure 9 shows the training progress.

## D.4  SOLVE PDEs

In this section, we utilize the LagrangeEmbedding-based network to address the following partial differential equations (PDEs):

$$\begin{cases} \Delta u + (u - \beta)^2 = (\alpha \cos x \sin y - 1)^2 + 1, & (x, y) \in \Omega; \\ u = \beta, & (x, y) \in \partial\Omega. \end{cases}$$

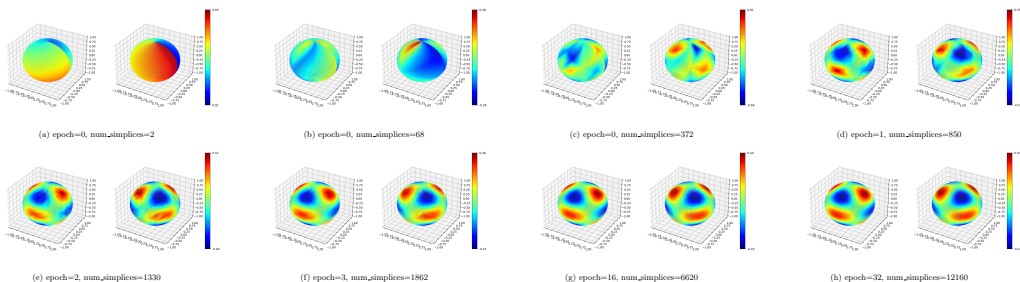

Figure 9: In each block, the left panel represents the real part of our model output, while the right panel represents the imaginary part of the model output.

Here, $\Omega = [0,1] \times [0,1]$. We construct a dataset that takes $(\alpha, \beta)$ as input data and assigns the corresponding numerical solution of the PDEs as the target output. This dataset comprises 12,000 examples, with $\alpha$ randomly selected from the distribution $U(-\pi/2, \pi/2)$ and $\beta$ randomly chosen from the distribution $U(0, 2\pi)$. We then split the dataset into two parts: 10,000 for training and 2,000 for testing. Figure 10 illustrates how well the network predicts the exact solution.

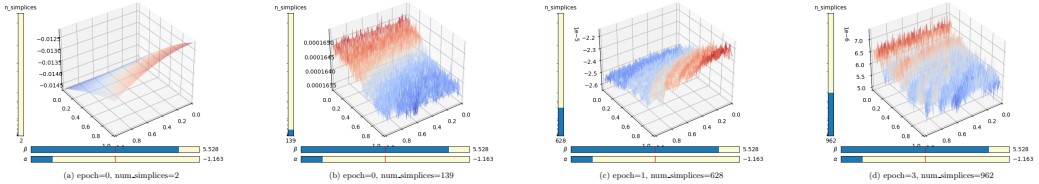

Figure 10: Residual - The gap between the exact solution and the model output.

## E    ABLATION STUDIES

In Sections 3 and 5, when we mentioned that "LagrangeEmbedding can be comparable to neural network-based encoders," we intended to convey that "the LagrangeEmbedding-based network performs equally well or even better than models of similar scale." To avoid any potential misunderstanding, we have conducted additional experiments for image classification tasks.

Table 2 presents the results of these experiments, which were carried out on MNIST, Fashion-MNIST, CIFAR-10, and CIFAR-100 datasets. The 4-layer model instances and training configurations used in these experiments were sourced from the PyTorch tutorial page (`https://github.com/pytorch/examples/tree/main/mnist`). In their setup, the AdaDelta optimizer was employed to minimize cross-entropy loss, starting with a learning rate of 1.0 and reducing it by a factor of 0.7 after each epoch. The batch size was set to 64, and the total training epochs were set to 14.

In Table 2, when the "EMBED" column is marked as False, it indicates that we trained the entire CNN. When "EMBED" is marked as True, we froze the CNN (except for the last linear layer) as a preprocessing layer and inserted our parameter-free LagrangeEmbedding before the final linear layer. We then trained only the final linear layer. If the "DATA AUG" is marked as True, it signifies the use of random horizontal flip as a data augmentation technique. The "INIT TIME" column reports the wall-clock time taken for initializing our LagrangeEmbedding on an RTX 2080ti GPU.

In the case of using LagrangeEmbedding, to improve time efficiency and extract non-low-level features from large raw data, here we employ LagrangeEmbedding as a low-rank mapping. We decompose the $d$-dimensional input space into the product of $d$ one-dimensional spaces and apply

Table 2: Ablation studies.

| Dataset | EMBED | DATA AUG | ACC@1 | INIT TIME |
|---|---|---|---|---|
| MNIST | ✗ | ✗ | 99.14% | N/A |
| | ✓ | ✗ | 99.17% | 1.55s |
| Fashion MNIST | ✗ | ✗ | 92.47% | N/A |
| | ✓ | ✗ | 92.56% | 1.55s |
| Fashion MNIST | ✗ | ✓ | 92.14% | N/A |
| | ✓ | ✓ | 92.14% | 2.08s |
| CIFAR-10 | ✗ | ✗ | 71.43% | N/A |
| | ✓ | ✗ | 71.54% | 1.83s |
| CIFAR-10 | ✗ | ✓ | 71.93% | N/A |
| | ✓ | ✓ | 72.09% | 2.29s |
| CIFAR-100 | ✗ | ✗ | 36.50% | N/A |
| | ✓ | ✗ | 36.78% | 1.84s |
| CIFAR-100 | ✗ | ✓ | 36.58% | N/A |
| | ✓ | ✓ | 36.97% | 2.30s |

LagrangeEmbedding in each one-dimensional space:

Original LagrangeEmbedding : $\mathbb{R}^d \to [0, 1]^n$

$$\boldsymbol{x} \mapsto (\mathcal{L}_1(\boldsymbol{x}), \cdots, \mathcal{L}_n(\boldsymbol{x}))$$

Low Rank LagrangeEmbedding : $\mathbb{R} \times \cdots \times \mathbb{R} \to [0, 1]^{nd}$

$$\boldsymbol{x} \mapsto (\mathcal{L}_1(\boldsymbol{x}_1), \cdots, \mathcal{L}_n(\boldsymbol{x}_1), \cdots, \mathcal{L}_1(\boldsymbol{x}_d), \cdots, \mathcal{L}_n(\boldsymbol{x}_d))$$

This approach partitions the high-dimensional input space of LagrangeEmbedding to reduce computational costs and accelerate the initialization process.

In these experiments, the PyTorch model instance has the same size as the LagrangeEmbedding-based network. However, while the former trains the entire model, the latter only trains the final linear layer. Notably, the latter performs equal time efficiency during inference, and outperforms in evaluation results. Although the LagrangeEmbedding-based network requires initialization before use, its actual initialization time is minimal.

## F COMPARISON WITH KERNEL METHODS

Our LagrangeEmbedding-based network and the kernel method share some high-level properties. For instance, both are linear models and serve as universal approximators. However, their underlying principles are different: kernel methods map low-dimensional linearly inseparable data to high-dimensional and linearly separable data, whereas our method only maps weak-correlated data to be linearly separable. Specifically, if two input data are very close, the inner product of their projections will be close to 1, but if their similarity exceeds the threshold (i.e., longer than one simplex), their projections will be orthogonal. From Figure 1, we have:

$$\text{LagrangeEmbedding}\left(\frac{1}{3}(p^{(1)} + p^{(2)} + p^{(3)})\right) = \left(0, \frac{1}{3}, \frac{1}{3}, \frac{1}{3}, 0, 0, 0, 0\right)$$

$$\text{LagrangeEmbedding}\left(\frac{1}{3}(p^{(2)} + p^{(3)} + p^{(4)})\right) = \left(0, 0, \frac{1}{3}, \frac{1}{3}, \frac{1}{3}, 0, 0, 0\right)$$

$$\text{LagrangeEmbedding}\left(\frac{1}{3}(p^{(0)} + p^{(5)} + p^{(6)})\right) = \left(\frac{1}{3}, 0, 0, 0, 0, \frac{1}{3}, \frac{1}{3}, 0\right)$$

The first point $\frac{1}{3}(p^{(1)} + p^{(2)} + p^{(3)}) \in \triangle p^{(1)}p^{(2)}p^{(3)}$ and the second point $\frac{1}{3}(p^{(2)} + p^{(3)} + p^{(4)}) \in \triangle p^{(2)}p^{(3)}p^{(4)}$ belong to adjacent triangles, so their LagrangeEmbedding projections are close, and

the inner product is close to 1. However, the first point $\frac{1}{3}(p^{(1)} + p^{(2)} + p^{(3)}) \in \triangle p^{(1)}p^{(2)}p^{(3)}$ and the third point $\frac{1}{3}(p^{(0)} + p^{(5)} + p^{(6)}) \in \triangle p^{(0)}p^{(5)}p^{(6)}$ belong to distant triangles, so their projections are orthogonal.

LagrangeEmbedding has two characteristics:

1. The inner product of two data in the input space does not necessarily correlate with the inner product in the projection space. In the projection space, their inner product is solely determined by the multi-scale mesh, and the mesh structure is determined by learning the data distribution in the input space. Therefore, for LagrangeEmbedding, two data with inner products close to 1 in the input space may potentially be orthogonal in the projection space.

2. The LagrangeEmbedding-based network can have unlimited width if we continuously increase its degrees of freedom.

Finally, LagrangeEmbedding has a cost-efficient solution for extracting features from high-dimensional raw data (see Appendix E, Low-Rank LagrangeEmbedding). Furthermore, LagrangeEmbedding can be used independently or as a parameter-free module inserted into a neural network to enhance the performance (see Appendix E, Table 2). Therefore, we believe our approach holds advantages over (Lee et al., 2017; Matthews et al., 2018; Kapoor et al., 2021).

