# OpenReview forum: "Is Training Necessary for Representation Learning"
_ICLR.cc/2024/Conference — Submitted to ICLR 2024_

### Official Review · Reviewer_YmUG · 2023-10-29

**Soundness:** 2 fair
**Presentation:** 4 excellent
**Contribution:** 3 good
**Rating:** 6
**Confidence:** 3

**Summary:**

The paper presents LagrangeEmbedding, a method for obtaining a data representation based on a first-order Lagrange basis. The authors present an algorithm for generating a mesh on the input space and show how to form the Lagrange basis in an efficient way using parallelization. The effectiveness of the method was demonstrated on several datasets, mostly in low-dimensional settings.

**Strengths:**

* A novel method for obtaining representations in an unsupervised that doesn't require any learning.
* The paper is written well and clearly. The visual illustrations are good and help to convey the main points in the paper.
* The method is grounded by theoretical justifications, such as the universal approximation, which adds a nice flavor to the paper.
* Code was provided and the results seems to be reproducible.

**Weaknesses:**

* The main limitation of the proposed approach, in my opinion, is that it misses the goal of the paper (or at least the one that was presented). I acknowledge that in some cases, mainly in low-dimensional settings or on simple datasets (e.g., MNIST) it works fine. However, the main goal of learning transferrable representations is from big, complex, and high-dimensional datasets. Currently, this approach doesn't fit these types of data, and I am skeptical if it will ever will as FEM is not a new idea. Hence, at least currently, I do not think this paper can make an impact as the authors imply in the paper. Nevertheless, I do appreciate novel and non-standard directions, even if they are not mature yet, and I give the paper credit for that.
* Another possible limitation relates to the fact that the method doesn't have any learnable parameters. Often there is a domain shift between the dataset for learning representations and the dataset of interest, or between the training set and test set. Standard NN-based approaches can work well in such cases (depending on the magnitude of the shift) or can be adjusted to these shifts by fine-tuning the feature extractor, for example. Yet, as the proposed method heavily relied on the algorithm for obtaining a multiscale mesh based on the training set, it is not clear how it will work in such cases.
* I expected to see a broader reference to kernel methods, but the paper seems to miss this related research direction entirely. Kernels also form a basis function and are also universal approximators, and perhaps there is some connection to the Lagrange basis function. More importantly, I find two studies particularly relevant (to address and compare to). First, the line of research on infinite-width NNs [1, 2] which also hinges on inner products in the input space. Second, the method presented in [3] also suggests to use simplices for approximating the full data kernel. When does the proposed approach preferred over these modeling choices?
* Regarding the experiments, it is not clear what is the test performance of the proposed approach on MNIST. If it is 97.25% as implied in the text, how is that equivalent to 6-layer CNN when even a standard LeNet reaches ~99.3% accuracy?
* In my opinion, some information is missing. Specifically,
  * A reference (or a proof) for the two properties of the Lagrange basis function in Section 2.
  * Intuition on the dimensionality reduction technique in section 3.2.1.
  * Why is the following true $(n_t/d!)^{1/d} = \mathcal{O}(h^{-1})$?

[1]  Lee, J., Bahri, Y., Novak, R., Schoenholz, S. S., Pennington, J., & Sohl-Dickstein, J. (2018, February). Deep Neural Networks as Gaussian Processes. In International Conference on Learning Representations.
[2]  Matthews, A. G. D. G., Hron, J., Rowland, M., Turner, R. E., & Ghahramani, Z. (2018, February). Gaussian Process Behaviour in Wide Deep Neural Networks. In International Conference on Learning Representations.
[3] Kapoor, S., Finzi, M., Wang, K. A., & Wilson, A. G. G. (2021, July). Skiing on simplices: Kernel interpolation on the permutohedral lattice for scalable gaussian processes. In International Conference on Machine Learning (pp. 5279-5289). PMLR.

**Questions:**

* At the beginning of Section 2.1, what is $m$, did you mean $N-1$?
* Perhaps I didn't understand something, but to me the definition of the Lagrange function $\mathcal{L}_i$ seems to be the indicator function and not a linear function picking at $p^{(i)}$.

---

> ### Author Response · Authors · 2023-11-20
> **Reply to Reviewer YmUG (W1, Part 1/2)**
>
> >**W1: I acknowledge that in some cases, mainly in low-dimensional settings or on simple datasets (e.g., MNIST) it works fine. However, the main goal of learning transferrable representations is from big, complex, and high-dimensional datasets. Currently, this approach doesn't fit these types of data.**
>
> We sincerely appreciate your thorough review of our paper and the concerns you raised regarding our experiments. Here is our response to your questions:
>
> Theoretically, as long as there are enough computational resources, LagrangeEmbedding can extract features from high-dimensional data. According to the important conclusion of FEM and the approximation theory, the linear combination of Lagrange basis functions must adhere to the **error-bound formula** Eqn (1):
> $$\\frac{1}{m} \\sum_{i=0}^{m-1} |f(x^{(i)};\\theta)-y^{(i)}|^2 = O(h) = O(n_t^{-1/d}) = O(n^{-1/d})$$
> The formula indicates that for $d$-dimensional inputs and MSE loss function, the test $MSE$ of LagrangeEmbedding-based models will decrease to $2^{−1/d} ∗ MSE$ when we double the number of parameters in last linear layer. Therefore, **we can continuously improve the model performance by increasing the number of parameters in the linear layer**. We have experimentally validated this in Section 3.1.3.
>
> For the applied scenario, when computational resources are limited, two solutions are available:
> 1. Design more efficient dimensionality reduction tools as preprocessing layers for LagrangeEmbedding-based networks.
> 2. Utilize LagrangeEmbedding as a low-rank mapping, decomposing the $d$-dimensional input space into the product of $d$ one-dimensional spaces and applying LagrangeEmbedding in each one-dimensional space​​:
>
> $$\\mathrm{Original~LagrangeEmbedding}: \\mathbb{R}^d \\rightarrow [0, 1]^n $$
>
> $$~~~~~~~~~~~~~~~~~~~~~~~~~~~~~~~~~~~~~~~~~~~~~~~~~~~~~~~~\\boldsymbol{x} \\mapsto (\\mathcal{L}_1(\\boldsymbol{x}), \\cdots, \\mathcal{L}_n(\\boldsymbol{x}))$$
>
> $$\\mathrm{LowRank~LagrangeEmbedding}: \\mathbb{R} \\times \\cdots \\times \\mathbb{R} \\rightarrow [0, 1]^{nd}$$
>
> $$~~~~~~~~~~~~~~~~~~~~~~~~~~~~~~~~~~~~~~~~~~~~~~~~~~~~~~~~~~~~~~~~~~~~~~~~~~~\\boldsymbol{x} \\mapsto (\\mathcal{L}_1(\\boldsymbol{x}_1), \\cdots, \\mathcal{L}_n(\\boldsymbol{x}_1), \\cdots, \\mathcal{L}_1(\\boldsymbol{x}_d), \\cdots, \\mathcal{L}_n(\\boldsymbol{x}_d))$$
>
> To assess the effectiveness of the second approach, we conducted ablation studies. **Table R2-1** (see RtZ7-W1) shows that using a 4-layer LeNet (RtZ7's instance https://github.com/pytorch/examples/tree/main/mnist) as a frozen preprocessing layer, the LagrangeEmbedding-based network achieved higher test accuracy than pre-trained models. Hence, LagrangeEmbedding can extract better features from MNIST, Fashion MNIST, CIFAR-10, and CIFAR-100. This approach has been added to Appendix E. Reviewers can reproduce our experimental results by running **run.sh** in the revised supplementary material.
>
> Finally, due to the limited time of the rebuttal phase, we have not yet tuned the training hyper-parameters on ImageNet and COCO-related experiments. Once we obtain SOTA results, we will promptly provide the corresponding code in our open-source project and publish it as a new paper. We again express our gratitude for your valuable feedback and look forward to your further review.

---

> ### Author Response · Authors · 2023-11-20
> **Reply to Reviewer YmUG (W1, Part 2/2)**
>
> >**W1: FEM is not a new idea. Hence, at least currently, I do not think this paper can make an impact as the authors imply in the paper.**
>
> It's true that FEM is not a new concept. However, the current FEM-related papers (e.g., [4][5]) are using neural networks to simulate the FEM without implementing FEM explicitly. Here are three pieces of evidence:
> 1. A hallmark of successfully constructing FEM is that the model must adhere to the **error-bound formula** Eqn. (1). This formula indicates that as long as the model's scale is sufficiently large, its performance can reach the theoretical maximum. However, in the FEM-related papers published so far, no one has demonstrated that they have achieved this. In contrast, Our experiments in Section 3.1.3 illustrate that we did it.
> 2. In FEM, the support of the basis functions is required to be independent of targets. However, In these papers, they train encoders, which means that the support of the basis functions is altered during training, which is not permitted in standard FEM. In contrast, we derive the fixed basis functions by mathematical deduction to make the train-free encoder.
> 3. FEM requires continuous variables as input, but images and text belong to discrete distributions. Therefore, using FEM directly for classification tasks cannot guarantee the **error-bound formula**, and will result in very poor results. In contrast, we design a multi-scale domain decomposition algorithm, as it ensures that FEM can be applied to neural network modeling.
>
> FEM-based software has been successfully used in various advanced engineering fields as the most outstanding algorithm in numerical simulation. As long as the computing resources are sufficient, there is almost no upper limit to the performance of this method. The current implementation of FEM is mature on CPU clusters but challenging on GPU-based parallel platforms. We precisely implement the Lagrange basis functions on PyTorch as one of the main contributions of our paper.

---

> ### Author Response · Authors · 2023-11-20
> **Reply to Reviewer YmUG (W2)**
>
> >**W2: Standard NN-based approaches can adjust the shifts between the dataset for learning representations and the dataset of interest by fine-tuning the feature extractor. Yet, as the proposed method heavily relied on the algorithm for obtaining a multiscale mesh based on the training set, it is not clear how it will work in such cases.**
>
> 1. Our LagrangeEmbedding is a universal encoder that can be applied to multiple tasks without fine-tuning. In contrast with standard transfer learning, LagrangeEmbedding is train-free, ensuring lossless memory and avoiding catastrophic forgetting. Admittedly, due to its generality, LagrangeEmbedding-based networks may not outperform task-specific neural networks in terms of accuracy for a particular recognition task. However, our models adhere to the **error-bound formula**, meaning that we can reduce this performance gap by increasing the scale of LagrangeEmbedding.
> 2. Our algorithm for generating multiscale meshes is designed to ensure that LagrangeEmbedding-based networks satisfy the **error-bound formula**. This algorithm determines the size of LagrangeEmbedding and the shape of its Lagrange basis functions.

---

> ### Author Response · Authors · 2023-11-20
> **Reply to Reviewer YmUG (W3)**
>
> >**W3: I expected to see a broader reference to kernel methods, but the paper seems to miss this related research direction entirely.**
>
> Thank you for your comments. We have added Appendix F to our paper to discuss the differences between LagrangeEmbedding and recommended papers [1][2][3]. Our LagrangeEmbedding-based network and kernel methods are both linear models. However, their underlying principles are different: kernel methods map low-dimensional linearly inseparable data to high-dimensional and linearly separable data, whereas our method only maps weak-correlated data to be linearly separable. Specifically, if two input data are very close, the inner product of their projections will be close to 1, but if their similarity exceeds the threshold (i.e., longer than one simplex), their projections will be orthogonal. From Figure 1 of our paper:
>
> $$\mathrm{Projection}(\frac{1}{3} (p^{(1)} + p^{(2)} + p^{(3)})) = [0, \frac{1}{3}, \frac{1}{3}, \frac{1}{3}, 0, 0, 0, 0]$$
>
> $$\mathrm{Projection}(\frac{1}{3} (p^{(2)} + p^{(3)} + p^{(4)})) = [0, 0, \frac{1}{3}, \frac{1}{3}, \frac{1}{3}, 0, 0, 0]$$
>
> $$\mathrm{Projection}(\frac{1}{3} (p^{(0)} + p^{(5)} + p^{(6)})) = [\frac{1}{3}, 0, 0, 0, 0, \frac{1}{3}, \frac{1}{3}, 0]$$
>
> The first point $\frac{1}{3} (p^{(1)} + p^{(2)} + p^{(3)}) \in \bigtriangleup p^{(1)}p^{(2)}p^{(3)}$ and the second point $\frac{1}{3} (p^{(2)} + p^{(3)} + p^{(4)}) \in \bigtriangleup p^{(2)}p^{(3)}p^{(4)}$ belong to adjacent triangles, so their LagrangeEmbedding projections are close, and the inner product is close to 1. However, the first point $\frac{1}{3} (p^{(1)} + p^{(2)} + p^{(3)}) \in \bigtriangleup p^{(1)}p^{(2)}p^{(3)}$ and the third point $\frac{1}{3} (p^{(0)} + p^{(5)} + p^{(6)}) \in \bigtriangleup p^{(0)}p^{(5)}p^{(6)}$ belong to distant triangles, so their projections are orthogonal.
>
> LagrangeEmbedding has two characteristics:
> 1. The inner product of two data in the input space does not necessarily correlate with the inner product in the projection space. In the projection space, their inner product is solely determined by the multi-scale mesh, and the mesh structure is determined by learning the data distribution in the input space. Therefore, for LagrangeEmbedding, two data with inner products close to 1 in the input space may potentially be orthogonal in the projection space.
> 2. The LagrangeEmbedding-based network can have unlimited width if we continuously increase its degrees of freedom.
>
> Finally, LagrangeEmbedding has a cost-efficient solution for extracting features from high-dimensional raw data (see W1, Part 1, **Low-Rank LagrangeEmbedding**). Furthermore, LagrangeEmbedding can be used independently or as a parameter-free module inserted into a neural network to enhance the performance (see RtZ7-W1 **Table R2-1**). Therefore, we believe our approach holds advantages over [1][2][3].

---

> ### Author Response · Authors · 2023-11-20
> **Reply to Reviewer YmUG (W4)**
>
> >**W4: Regarding the experiments, it is not clear what is the test performance of the proposed approach on MNIST. If it is 97.25% as implied in the text, how is that equivalent to 6-layer CNN when even a standard LeNet reaches ~99.3% accuracy?**
>
> Thank you for your feedback. Several LeNet models (from GitHub and PyTorch tutorial https://github.com/pytorch/examples/blob/main/mnist/main.py) achieve ~99.1% test accuracy on MNIST and have parameter counts ranging from 0.8 million to 1.2 million. In contrast, our classifier contains only 0.09 million model parameters. For a fair comparison, our LagrangeEmbedding-based model maintains its advantage in the same parameter counts. In **Table R2-1** (see RtZ7-W1), we show the better performance of LagrangeEmbedding-based networks on other datasets. In Section 3.3, the LagrangeEmbedding-based network contains only 256 parameters but reaches 90% test accuracy on the text classification task, which is the same accuracy performance as the word2vec model (see the instance https://pytorch.org/tutorials/beginner/text_sentiment_ngrams_tutorial.html).

---

> ### Author Response · Authors · 2023-11-20
> **Reply to Reviewer YmUG (W5)**
>
> >**W5: In my opinion, some information is missing. Specifically, (1) A reference (or a proof) for the two properties of the Lagrange basis function in Section 2. (2) Intuition on the dimensionality reduction technique in section 3.2.1. (3) Why is $(n_t/d!)^{1/d} = O(h^{-1})$ true?**
>
> 1. We have now included relevant references in the revised paper, such as [6]. The property "Universal Approximation" is from approximation and linear interpolation theory, while the "Similarity Calculation" property is based on the theory of first-order linear finite elements.
> 2. The dimensionality reduction tool (the untrainable preprocessing layer) in Section 3.2.1 is not well-designed. Replacing it with random linear weighting for dimensionality reduction does not significantly impact the experimental results. Therefore, in Section 4, we stated that our future research direction focuses on constructing a better dimensionality reduction tool that compresses information without loss.
> 3. This is a conclusion from FEM. When $d=1,2$, the formula can be derived from the standard uniform mesh (see https://en.wikipedia.org/wiki/Triangular_tiling). In higher dimensions, it can be proven via mathematical induction.

---

> ### Author Response · Authors · 2023-11-20
> **Reply to Reviewer YmUG (Q1 & 2)**
>
> >**Q1: At the beginning of Section 2.1, what is $m$, did you mean $N-1$?**
>
> Thank you for correcting me. A clear statement would be: ``For any given simplex, select a subset $\{ (x^{(k_0)}, y^{(k_0)}), \cdots, (x^{(k_{m'-1})}, y^{(k_{m'-1})}) \}$ from the training set $\{ (x^{(0)}, y^{(0)}), \cdots, (x^{(m-1)}, y^{(m-1)}) \}$ where $m'$ is the cardinality of subset, $m$ is the cardinality of subset, and all subset elements reside within the given simplex."
>
> >**Q2: Perhaps I didn't understand something, but to me the definition of the Lagrange function $\mathcal{L}_i$ seems to be the indicator function and not a linear function picking at $p^{(i)}$.**
>
> The Lagrange basis function is globally continuous and piecewise linear, not an indicator function. Please refer to Figure 6 (Middle) in Appendix B, It's a function image demo for 2D Lagrange function $L_7$ and $L_{20}$. Note: unlike some kernel methods, the support of Lagrange bais are not isolated from each other.

---

> ### Author Response · Authors · 2023-11-20
> **Reply to Reviewer YmUG (References)**
>
> [1] Lee, J., Bahri, Y., Novak, R., Schoenholz, S. S., Pennington, J., & Sohl-Dickstein, J. (2018, February). Deep Neural Networks as Gaussian Processes. In International Conference on Learning Representations.
>
> [2] Matthews, A. G. D. G., Hron, J., Rowland, M., Turner, R. E., & Ghahramani, Z. (2018, February). Gaussian Process Behaviour in Wide Deep Neural Networks. In International Conference on Learning Representations.
>
> [3] Kapoor, S., Finzi, M., Wang, K. A., & Wilson, A. G. G. (2021, July). Skiing on simplices: Kernel interpolation on the permutohedral lattice for scalable gaussian processes. In International Conference on Machine Learning (pp. 5279-5289). PMLR.
>
> [4] Hashash, Y. M. A., Jung, S., & Ghaboussi, J. (2004). Numerical implementation of a neural network based material model in finite element analysis. International Journal for numerical methods in engineering, 59(7), 989-1005.
>
> [5] Javadi, A. A., Tan, T. P., & Zhang, M. (2003). Neural network for constitutive modelling in finite element analysis. Computer Assisted Mechanics and Engineering Sciences, 10(4), 523-530.
>
> [6] Zienkiewicz, O. C., Morgan, K., & Morgan, K. (2006). Finite elements and approximation. Courier Corporation.

---

> > ### Comment · Reviewer_YmUG · 2023-11-22
> > **Response to Authors**
> >
> > I would like to thank the authors for the answers and empirical evaluations following the comments made by myself and other reviewers. I have a follow-up question that also relates to my point on OOD data. You state that your model adheres to the error-bound formula, yet, and perhaps I misunderstood something, the algorithm for generating the mesh (Algo. 1) depends on the training data locations. Is that right? If indeed that is true, then some regions, especially those that are not located near the training data points, will be sparsely covered by the mesh (or will not be covered at all). Hence, there is another factor here besides the FEM, your algorithm, which may introduce another error. So, even if you increase the resolution of the mesh (which to my understanding will be denser only in regions of the training data), the proposed bound will not be correct, no?
> >
> > Also, regarding Q2 in my original comment. I am sorry for the confusion. What I meant is that on page 2, you define the Lagrange function in terms of the node only which can be perceived as an indicator function picking at the corresponding node. But now, after a second read, I understand what you meant.

---

> ### Author Response · Authors · 2023-11-22
> **Response to Reviewer YmUG**
>
> > I have a follow-up question that also relates to my point on OOD data. You state that your model adheres to the error-bound formula, yet, and perhaps I misunderstood something, the algorithm for generating the mesh (Algo. 1) depends on the training data locations. Is that right? If indeed that is true, then some regions, especially those that are not located near the training data points, will be sparsely covered by the mesh (or will not be covered at all). Hence, there is another factor here besides the FEM, your algorithm, which may introduce another error. So, even if you increase the resolution of the mesh (which to my understanding will be denser only in regions of the training data), the proposed bound will not be correct, no?
>
> We sincerely appreciate the reviewer's feedback.  As confirmed in Section 3.1.3, the LagrangeEmbedding-based network adheres to the error-bound formula of FEM and approximation theory on the test set. These experimental results demonstrate that our approach handles out-of-distribution (OOD) data well.
>
> Based on Figure 1 or Eqn (3), **if a test data point is located in any simplex of the mesh, LagrangeEmbedding will map it to a non-zero vector, whether it resides in a coarse or fine simplex.** Specifically, If a test data point is within a coarse simplex, there are none or rare training data points similar to this data, indicating lower confidence in predicting this data.
>
> Conversely, if the test data point is outside the mesh, LagrangeEmbedding will map it to a zero vector. To prevent this from happening, the initial mesh must be sufficiently large. As we introduced at the beginning of Alg 1, "Initial Node Matrix $P$ containing coordinates of $d + 1$ points forming a simplex covering all training raw data", it refers to the following large simplex (see `utils.py`/`utilsv2.py`) $T$ given by its vertices:
>
> $$V(T) = \\{ \boldsymbol{v}^{(j)} | v_k^{(j)} = 1_{j \neq k} [(1+\epsilon) \min_i x_k^{(i)} - \epsilon \max_i x_k^{(i)}] + 1_{j = k} [(2+3\epsilon) \max_i x_k^{(i)} - (1+3\epsilon) \min_i x_k^{(i)}], j=0, \cdots d-1; v_k^{(d)} = (1+\epsilon) \min_i x_k^{(i)} - \epsilon \max_i x_k^{(i)} \\}, \epsilon=0.1.$$
>
>
> In a particular scenario, if the empirical distribution of the training set and test set are significantly different in the boundary area, causing some test data points still lie outside the mesh. The error-bound formula Eqn (1) will still hold when such abnormal data are in the minority:
>
> $$\frac{1}{m} \sum_{i=0}^{m-1} |f(x^{(i)};\theta)-y^{(i)}|^2 =
> \frac{1}{m} \sum_{x^{(i)} \in T} |f(x^{(i)};\theta)-y^{(i)}|^2 + \frac{1}{m} \sum_{x^{(i)} \notin T} |f(x^{(i)};\theta)-y^{(i)}|^2 = \Theta (\frac{1}{m} \sum_{x^{(i)} \in T} |f(x^{(i)};\theta)-y^{(i)}|^2) = O(n^{-1/d})$$

---

### Official Review · Reviewer_RtZ7 · 2023-10-31

**Soundness:** 2 fair
**Presentation:** 2 fair
**Contribution:** 2 fair
**Rating:** 5
**Confidence:** 3

**Summary:**

The paper proposes a training-free approach for generating a feature vector, where each coordinate value corresponds to the output of a Lagrange basis function. The proposed method enjoys theoretical guarantees on its approximation error as a function of the number of parameters. The effectiveness of the resulting embedding is evaluated on fitting data drawn from known distributions, as well as classification/super-resolution on the MNIST dataset, and classification on AG News dataset.

**Strengths:**

- The method does not require training/backpropagation to generate input embeddings.
- The method is evaluated across multiple tasks from traditional data fitting, to computer vision and NLP tasks.
- The method enjoys theoretical bounds on the approximation error given the number of model parameters.
- Limitations are discussed

**Weaknesses:**

- The paper over-promises and under-delivers. Among other broad claims - e.g. first paragraph of the conclusion, ``unparalleled level of explainability” - the title itself “Is training necessary for representation learning” suggests that the proposed method can be comparable to training-based approaches such as neural networks. Yet, there exists few, if any, quantitative comparisons between the proposed method and neural network approaches, especially for the (toy) computer vision and NLP experiments.
- In fact, the basic 2-layer convolutional network, for instance taken from the PyTorch tutorial page (https://github.com/pytorch/examples/tree/main/mnist), already achieves 98% accuracy on MNIST in the first epoch (outperforming the proposed approach), which completes in under a minute on a CPU and presumably orders of magnitude faster on GPU.
- Sec 3.1.2 compares against neural networks when fitting distributions drawn from 2-dimensional distributions, but it is not stated what network parameters nor training parameters are used other than the fact that it is a MLP.
- It seems that in Table 1, Random Forest is already highly effective at achieving almost perfect R^2 scores, and performance on most of the distributions considered appears to have already saturated.
- How did the projection layer in Sec 3.2.1 arise? There is no explanation for why this specific projection equation was introduced, and while it claims to contain “no trainable model parameters”, it appears to require careful hand-crafting as well.
- Speed is touted as an advantage of the method, but there exists no wall-clock timing comparisons for computing the proposed embedding.


Minor comments
- Eqn (2) $y^{(j)}$ should be $y^{(i)}$ instead
- In Sec 3.3, does “the neural network” refer to the proposed method (i.e. typo)? If not, are there quantitative results and comparisons for the proposed method?
- Also in Appendix D.2., I assume “Remarkably, after just 4 epochs of training, the neural network outputs close approximate the target values” is also a typo?

**Questions:**

- Sec 3.3 - can you elaborate on how the pre-processing layer is implemented?

---

> ### Author Response · Authors · 2023-11-20
> **Reply to Reviewer RtZ7 (Part 1/3)**
>
> > **W1: The paper over-promises and under-delivers. Among other broad claims - e.g. first paragraph of the conclusion, ``unparalleled level of explainability” - the title itself “Is training necessary for representation learning” suggests that the proposed method can be comparable to training-based approaches such as neural networks. Yet, there exists few, if any, quantitative comparisons between the proposed method and neural network approaches, especially for the (toy) computer vision and NLP experiments.**
>
> 1. Thank you for your feedback. We place a strong emphasis on the explainability of our LagrangeEmbedding. As we demonstrated in Section 2 & 3.1.3, LagrangeEmbedding-based models strictly follow the complete FEM theory due to adhering to the **error-bound formula**:
> $$\frac{1}{m} \sum_{i=0}^{m-1} |f(x^{(i)};\theta)-y^{(i)}|^2 = O(h) = O(n_t^{-1/d}) = O(n^{-1/d})$$
> For $d$-dimensional inputs and MSE loss function, this formula shows that when we double the parameters counts of the last linear layer, the training error $MSE$ of LagrangeEmbedding-based models will get reduced to $2^{-1/d}*MSE$. **Therefore, we can continuously improve the model performance by increasing the number of parameters in the linear layer.**
>
> 2. In Sections 3 & 5, when we mentioned that "LagrangeEmbedding can be comparable to neural network-based encoders," we meant that "the LagrangeEmbedding-based network performs equally well or even better than models of similar scale." To avoid any misunderstanding, we have included additional experiments for image classification tasks:
>
> **Table R2 - 1**: Ablation studies. We present the experimental results for image classification tasks conducted on MNIST, FashionMNIST, CIFAR-10, and CIFAR-100 datasets. The model instances used in these experiments are obtained from the PyTorch tutorial page (https://github.com/pytorch/examples/tree/main/mnist). When the "LagrangeEmbed" column is marked as "False," it indicates that we trained the entire CNN. When "LagrangeEmbed" is marked as "True," we freeze the CNN (excluding the last linear layer) as a preprocessing layer and insert our parameter-free LagrangeEmbedding before the final linear layer, then only train the final linear layer. The "data_aug=True" signifies using random horizontal flip as a data augmentation technique.
> The "init time" column reports the wall-clock time taken for initializing our LagrangeEmbedding on an RTX 2080ti GPU. All training configurations are taken from the source PyTorch tutorial page, and all experiments can be reproduced by running `bash run.sh` in our revised supplementary material.
> | dataset  | LagrangeEmbed | data aug | test acc | init time |
> | -------- | ------------- | -------- | -------- | --------- |
> | minist       | False     | False    | 99.14%   | N/A       |
> | minist       | True      | False    | 99.17%   | 1.55s     |
> | fashionmnist | False     | False    | 92.47%   | N/A       |
> | fashionmnist | True      | False    | 92.56%   | 1.55s     |
> | fashionmnist | False     | True     | 92.14%   | N/A       |
> | fashionmnist | True      | True     | 92.14%   | 2.08s     |
> | cifar10      | False     | False    | 71.43%   | N/A       |
> | cifar10      | True      | False    | 71.54%   | 1.83s     |
> | cifar10      | False     | True     | 71.93%   | N/A       |
> | cifar10      | True      | True     | 72.09%   | 2.29s     |
> | cifar100     | False     | False    | 36.50%   | N/A       |
> | cifar100     | True      | False    | 36.78%   | 1.84s     |
> | cifar100     | False     | True     | 36.58%   | N/A       |
> | cifar100     | True      | True     | 36.97%   | 2.30s     |
>
> **Table R2-1** shows that LagrangeEmbedding can be inserted as a parameter-free module into a frozen model, where the improved model size remains almost unchanged, yet achieves better model performance. For text classification tasks, our LagrangeEmbedding-based network in Section 3.3 contains only 256 parameters but performs on par with the 6.13 million-parameter word2vec model.

---

> ### Author Response · Authors · 2023-11-20
> **Reply to Reviewer RtZ7 (Part 2/3)**
>
> >**W2: In fact, the basic 2-layer convolutional network, for instance taken from the PyTorch tutorial page (https://github.com/pytorch/examples/tree/main/mnist), already achieves 98% accuracy on MNIST in the first epoch (outperforming the proposed approach), which completes in under a minute on a CPU and presumably orders of magnitude faster on GPU.**
>
> Thank you to the reviewer for providing the model instance (https://github.com/pytorch/examples/tree/main/mnist). We checked it and found it has 11 times more parameters than our model used in Section 3.2. To evaluate the Lagrange-based network fairly, we revised the implementation of our multi-scale domain decomposition method to run on GPU and conducted the experiments as shown in **Table R2-1**.
>
> In these experiments, the PyTorch model instance has the same size as the LagrangeEmbedding-based network. However, while the former trains the entire model, the latter only trains the final linear layer. Notably, the latter performs equal time efficiency during inference, and outperforms in evaluation results. Although the LagrangeEmbedding-based network requires initialization before use, its actual initialization time is minimal.
>
> >**W3: Sec 3.1.2 compares against neural networks when fitting distributions drawn from 2-dimensional distributions, but it is not stated what network parameters nor training parameters are used other than the fact that it is a MLP.**
>
> Thank you for your feedback. The neural network we described in Figure 3 is a 5-layer MLP using the ReLU activation function. No matter how we increase the number of parameters of this MLP or tune the training parameter configuration, the MLP cannot fit the high-frequency area in dataset $\mathbb{C}$ well. This experimental phenomenon is consistent with the description in the paper [1].
>
> >**W4: It seems that in Table 1, Random Forest is already highly effective at achieving almost perfect $R^2$ scores, and performance on most of the distributions considered appears to have already saturated.**
>
> Yes. For data fitting tasks, random forests exhibit similar performance to our LagrangeEmbedding-based network and overcome the challenge of fitting multi-frequency data. However, unlike our LagrangeEmbedding-based network, in most cases, random forests are not suitable for image and text-related recognition tasks.
>
> >**W5: How did the projection layer in Sec 3.2.1 arise? There is no explanation for why this specific projection equation was introduced, and while it claims to contain “no trainable model parameters”, it appears to require careful hand-crafting as well.**
>
> We appreciate the feedback from the reviewer. The architecture of our LagrangeEmbedding-based network consists of a "preprocessing layer -> LagrangeEmbedding -> linear layer," where the static preprocessing layer serves as a dimensionality reduction tool, and the train-free LagrangeEmbedding acts as the projection layer.
>
> As we explained in the **Remark** of Section 3.2.1, the dimensionality reduction tool (the untrainable preprocessing layer) in Section 3.2.1 wasn't well-designed. Replacing it with random linear weighting for dimensionality reduction had minimal impact on the experimental results. To prevent any misunderstanding and highlight the advantages of LagrangeEmbedding, please refer to the preceding **Table R2-1**.
>
> >**W6: Speed is touted as an advantage of the method, but there exist no wall-clock timing comparisons for computing the proposed embedding.**
>
> We have updated our supplementary materials, and the multi-scale domain decomposition method can now be executed on GPU. In terms of image classification tasks, our model demonstrates similar time efficiency compared to other networks (refer to **Table R2-1** for details. To reproduce these results, please run `bash run.sh`).
>
> For text classification tasks, our model achieves faster performance, with an epoch taking 74.99 seconds (online training on RTX 2080Ti), compared to the word2vec model (https://pytorch.org/tutorials/beginner/text_sentiment_ngrams_tutorial.html), which requires 91.24 seconds per epoch (online training on RTX 2080Ti). To reproduce these results, please run `textv2.py` and `textv2_comparion.py`.

---

> ### Author Response · Authors · 2023-11-20
> **Reply to Reviewer RtZ7 (Part 3/3)**
>
> # Minor comments
>
> >**C1: Eqn (2) $y^{(j)}$ should be $y^{(i)}$ instead.**
>
> Thank you for reminding us of the typo!
>
> >**C2: In Sec 3.3, does “the neural network” refer to the proposed method (i.e. typo)? If not, are there quantitative results and comparisons for the proposed method?**
>
> It is not a typo. Here, "the neural network" refers to our LagrangeEmbedding-based network. **Our LagrangeEmbedding has only one hyperparameter: the degrees of freedom (dof).** When we set the dof to 64, the test accuracy on AG news dataset exceeds 90%. For lower dof, the test accuracy will decrease.
>
> >**C3: Also in Appendix D.2., I assume “Remarkably, after just 4 epochs of training, the neural network outputs close approximate the target values” is also a typo?**
>
> This is not a typo. The meaning of this sentence is that the LagrangeEmbedding-based network converged after the first epoch training and did not obtain more test accuracy in the following four epochs.
>
> # Questions
>
> >**Q1: Sec 3.3 - can you elaborate on how the pre-processing layer is implemented?**
>
> Yes, from our supplementary materials, the pre-processing layer outputs the TF [2] value where
> $$\\mathrm{TF} = \\frac{\\mathrm{number\\ of\\ times\\ the\\ term\\ appears\\ in\\ the\\ document}}{\\mathrm{total\\ number\\ of\\ terms\\ appears\\ in\\ the\\ document}}$$

---

> ### Author Response · Authors · 2023-11-20
> **Reply to Reviewer RtZ7 (Reference)**
>
> [1] Xu, Z. Q. J., Zhang, Y., & Xiao, Y. (2019). Training behavior of deep neural network in frequency domain. In Neural Information Processing: 26th International Conference, ICONIP 2019, Sydney, NSW, Australia, December 12–15, 2019, Proceedings, Part I 26 (pp. 264-274). Springer International Publishing.
>
> [2] Sparck Jones, K. (1972). A statistical interpretation of term specificity and its application in retrieval. Journal of documentation, 28(1), 11-21.

---

> > ### Comment · Reviewer_RtZ7 · 2023-11-22
> > **Thank you for the rebuttal**
> >
> > Thank you authors for the response. I still have some remaining concerns, especially regarding the main claim of the paper
> >
> > Regarding authors' latest experiments
> >
> > > **Therefore, we can continuously improve the model performance by increasing the number of parameters in the linear layer.**
> >
> > This claim needs to be empirically backed, since it is expected that generalization performance saturates or decreases after some point. Model performance, or in general how good the representations are, should not be measured on the training set (as done in the theory, since NNs can overfit to even random finite training data given enough parameters), but on unseen data. The latter case needs to be experimentally determined, since the theory does not bound this generalization error.
> >
> > > **Table R2 - 1: Ablation studies**
> >
> > Thank you for the additional comparison. The additional results indeed reflect that the proposed method can perform similarity to the MNIST CNN architecture. However, CIFAR-10 etc. are problems which have already been solved by training-based representation learning methods. The proposed method still yields poor performance on such datasets (and it is not clear whether they can even scale especially regarding generalization, as noted by previous comment), which suggests the proposed method is still far from competing with learnt representations especially since the overall computation time is still similar.
> >
> > A minor comment: It might be confusing to readers to use "the neural network" interchangeably with the proposed network, since it appears that NNs in general are comparison baselines for the proposed approach.
> >
> > I raised my score to 5, since the experiments conclude that the proposed method performs comparably to toy NNs, and better than random embeddings. I did not raise it further because
> >
> > - The proposed method has not be shown to be able to scale, in terms of generalization, with number of parameters to better close the gap to larger NNs even on the toy tasks of CIFAR-10/100 (see first part of this reply)
> > - As such, the claim implied by the title seem to contradict the experiment results, since the proposed method performs poorly on tasks already solved by NNs
> > - On the toy vision models, the method does not seem to provide any significant advantage in terms of speed.
> > - The claim that the method provides "unparalleled level of explainability" is not backed by any evidence nor comparisons.

---

> ### Author Response · Authors · 2023-11-22
> **Response to Reviewer RtZ7**
>
> > This claim needs to be empirically backed, since it is expected that generalization performance saturates or decreases after some point. Model performance, or in general how good the representations are, should not be measured on the training set, but on unseen data. The latter case needs to be experimentally determined, since the theory does not bound this generalization error.
>
> Thank you for the valuable insights. We have the same point: generally, the relationship between model capacity and its performance exhibits Marginal Utility. People cannot continuously improve their model performance on the test set by increasing the model size without limits.
>
> However, this relationship is explicit to our model, and the optimal capacity (the point of generalization performance saturation) is related to the number of training examples: the relationship between the capacity of the LagrangeEmbedding-based network and its test error adheres to the **error-bound formula** until it reaches optimal capacity. To substantiate this claim, in Section 3.1.3, we used a more precise metric, the $l^2$ error, to assess the model's generalization performance:
>
> $$l^2 = E_{(x,y) \sim p_{data}} [|f(x;\theta) - y|^2] = \int |f(x;\theta) - y|^2 \ \mathrm{d}x.$$
>
> Here, we evaluate the model's performance on the given data-generating distribution $p_{data}$ where the $l^2$ error is a numerical integral over the whole input space. The entire input space contains not only the test set but also more unseen data. Therefore, if the $l^2$ error is small, the test MSE will be smaller.
>
> In Section 3.1.3, we conducted experiments on one-dimensional and two-dimensional data fitting tasks. The datasets consisted of 1000 training examples and 200 test examples for the one-dimensional case and 7500 training examples and 1500 test examples for the two-dimensional case. These datasets are sufficient for low-dimensional regression tasks. Figure 4 shows, that we observed that "the relationship between the number of linear layer parameters and performance on test set adheres the **error-bound formula**." Furthermore, we also observed this phenomenon in the MNIST super-resolution task.
>
> > The additional results indeed reflect that the proposed method can perform similarity to the MNIST CNN architecture. However, CIFAR-10 etc. are problems which have already been solved by training-based representation learning methods. The proposed method still yields poor performance on such datasets (and it is not clear whether they can even scale especially regarding generalization, as noted by previous comment), which suggests the proposed method is still far from competing with learnt representations especially since the overall computation time is still similar.
>
> Thank you for the suggestions, and we will do more experiments in our future work. The ablation experiments in **Table R2-1** demonstrate that we can improve the performance of a given model by using LagrangeEmbedding, which validates that LagrangeEmbedding indeed extracts better features. **Table R2-1** shows low test accuracy because the given baseline model is a simple 4-layer CNN.
>
> ---
>
> Finally, we regret that we couldn't provide more extensive experimental results before the rebuttal deadline, especially the experiments for large models and large datasets. Here is what we are currently working on:
>
> 1. The explainability of LagrangeEmbedding is manifested in its adherence to the **error-bound formula**, which we have experimentally proved in data fitting and image super-resolution tasks. We are in the process of validating this on better models and bigger datasets.
> 2. The **error-bound formula** of FEM requires the loss function to be MSE, while classification tasks generally use cross-entropy as the loss function. Although our experiments in Table 1 and **Table R2-1** demonstrate the effectiveness of LagrangeEmbedding, we are still studying the theoretical principles in an exploratory stage.

---

### Official Review · Reviewer_P3jk · 2023-11-03

**Soundness:** 3 good
**Presentation:** 2 fair
**Contribution:** 3 good
**Rating:** 6
**Confidence:** 3

**Summary:**

In this paper, the author proposed a feature extraction method termed LagrangeEmbedding, which can extract features from simple image and text datasets. LagrangeEmbedding fits a function with many piecewise linears. The proposed method is validated with regressor and classification tasks.

Overall, the ideal is novel, which can inspire further development of unsupervised representation learning. However, the related works that are closely related to the thinking of LagrangeEmbedding should be given. The proposed method seems to only work on simple datasets. What's more, the performance comparison is not provided.

**Strengths:**

1. The idea is novel. It provides a novel perspective for unsupervised representation learning.
2. Some detailed examples and analyses are provided.

**Weaknesses:**

1. The proposed method seems to only work on some toy tasks.
2. The related work sections or some closely related works are not provided.
3. The proposed method is only validated on simple image and text datasets. The comparison results with SOTA methods are not given. Even the proposed method achieves lower accuracy than SOTA methods. The comparison experiment with SOTA methods can assist the reader in finding the gap between the proposed and SOTA methods.
4. The proposed method only runs in a non-parallel manner, as mentioned in the future directions section.

**Questions:**

1. It seems that the proposed method can only extract low-level features, unlike the deep learning-based methods. The extracted features seem only suitable for toy tasks. Does the proposed method can extract non-low-level features?
The author is suggested to add some analysis and discussion.
2. How can we extend the proposed method for complex tasks in actual situations? The author is suggested to add some discussion.
3. I have not seen the author mention some closely related works. Is the proposed method totally original? If not, please provide the detailed related works and the difference between the proposed method and the related works.
4. In section 2.1,  the definition of m in x^{(m)} is not given. What's the difference bettween the x^{(N-1)} and  x^{(m)} ?

Other suggestions:
a） In Eqn(2), the ‘i’ is suggested to be replaced with 'n';
b） ”given function F (x) to be fitted”  ->``given function F (x) to be fitted”
c） The definition of SVR is not given.

---

> ### Author Response · Authors · 2023-11-20
> **Reply to Reviewer P3jk (Part 1/2)**
>
> >**W1: The proposed method seems to only work on some toy tasks.**
>
> We sincerely appreciate the reviewer's feedback. We have now updated the GPU implementation of the multi-scale domain decomposition method in the supplementary materials. Additionally, for other datasets or more complex tasks, we have proposed several approaches and conducted further experiments to analyze their impact. The experimental results demonstrate that our LagrangeEmbedding is also fast and effective for other datasets (Please refer to our first response to YmUG-W1 and RtZ7-W1 for details).
>
> Due to the limited time of the rebuttal phase, we have not yet tuned the training hyper-parameters on ImageNet and COCO-related experiments. Once we obtain SOTA results, we will promptly provide the corresponding code in our open-source project and publish it as a new paper. We again express our gratitude for your valuable feedback and look forward to your further review.
>
> >**W2: The related work sections or some closely related works are not provided.**
>
> Thank you for the reviewer's reminder. Our LagrangeEmbedding encoder is original. Unlike other other FEM-related models, it is an explicit implementation of FEM rather than an imitation, and it is the first model that satisfies the FEM **error-bound formula**. Then, our approach differs from kernel methods, it maps two raw data to orthogonal vectors only when the two raw data have significant differences. We have included Appendix F in the revised paper to discuss the distinctions between LagrangeEmbedding-based networks and kernel methods.
>
> >**W3: The proposed method is only validated on simple image and text datasets. The comparison results with SOTA methods are not given.**
>
> Thank you for your feedback. Admittedly, due to the train-free property and generality of LagrangeEmbedding, LagrangeEmbedding-based networks may not outperform task-specific neural networks in terms of accuracy for a particular recognition task. However, our LagrangeEmbedding, can be inserted into any frozen neural network as a plug-and-play parameter-free module (Please refer to **Table R2-1** in our response to RtZ7-W1). It barely changes the model scale, only requires training the final linear layer, but enhances the model's performance. These experiments indicate that LagrangeEmbedding has the potential to help models achieve SOTA results.
>
> >**W4: The proposed method only runs in a non-parallel manner, as mentioned in the future directions section.**
>
> We have recently upgraded the GPU implementation of the multi-scale domain decomposition method in the revised supplementary materials. It is available to run the whole data pipeline of LagrangeEmbedding-based networks in GPU parallel mode.

---

> ### Author Response · Authors · 2023-11-20
> **Reply to Reviewer P3jk (Part 2/2)**
>
> >**Q1: It seems that the proposed method can only extract low-level features, unlike the deep learning-based methods. The extracted features seem only suitable for toy tasks. Does the proposed method can extract non-low-level features? The author is suggested to add some analysis and discussion.**
>
> Thank you for your feedback. To extract non-low-level features from large raw data, we can utilize LagrangeEmbedding as a low-rank mapping, decomposing the $d$-dimensional input space into the product of $d$ one-dimensional spaces and applying LagrangeEmbedding in each one-dimensional space​​:
> $$\\mathrm{Original~LagrangeEmbedding}: \\mathbb{R}^d \\rightarrow [0, 1]^n $$
>
> $$~~~~~~~~~~~~~~~~~~~~~~~~~~~~~~~~~~~~~~~~~~~~~~~~~~~~~~~~\\boldsymbol{x} \\mapsto (\\mathcal{L}_1(\\boldsymbol{x}), \\cdots, \\mathcal{L}_n(\\boldsymbol{x}))$$
>
> $$\\mathrm{LowRank~LagrangeEmbedding}: \\mathbb{R} \\times \\cdots \\times \\mathbb{R} \\rightarrow [0, 1]^{nd}$$
>
> $$~~~~~~~~~~~~~~~~~~~~~~~~~~~~~~~~~~~~~~~~~~~~~~~~~~~~~~~~~~~~~~~~~~~~~~~~~~~\\boldsymbol{x} \\mapsto (\\mathcal{L}_1(\\boldsymbol{x}_1), \\cdots, \\mathcal{L}_n(\\boldsymbol{x}_1), \\cdots, \\mathcal{L}_1(\\boldsymbol{x}_d), \\cdots, \\mathcal{L}_n(\\boldsymbol{x}_d))$$
> The ablation studies in **Table R2-1** (see RtZ7-W1) demonstrate that this approach is indeed effective. It successfully reduces the 12,544-dimensional features (the projection dimension of a 4-layer LeNet) to 2 * 12,544 one-dimensional features.
>
> >**Q2: How can we extend the proposed method for complex tasks in actual situations? The author is suggested to add some discussion.**
>
> If users have sufficient computing resources, they can address this issue by increasing the scale of LagrangeEmbedding. As LagrangeEmbedding-based networks adhere to the error-bound formula, **we can continuously improve the model performance by increasing the number of parameters in the linear layer**. For users with limited computational resources, we can consider the solution described in the previous question by partitioning the high-dimensional input space to reduce computational costs.
>
> >**Q3: Is the proposed method totally original?**
>
> Yes, our multi-scale domain decomposition method (DDM) and the construction method of the Lagrange basis are totally original. Specifically, we construct the LagrangeEmbedding architecture by using Eqn (3) and initialize LagrangeEmbedding size via the multi-scale domain decomposition method. Unlike existing FEM-related neural networks [1][2], we have implemented FEM explicitly, rather than imitating it. The evidence lies in the fact that our model adheres to the **error-bound formula**.
>
> >**Q4: In section 2.1, the definition of $m$ in $x^{(m)}$ is not given. What's the difference bettween the $x^{(N-1)}$ and $x^{(m)}$?**
>
> Thanks to the reviewer's reminder. Here, $m$ is the cardinality of the training set, and $N$ is the cardinality of the subset. A clear statement would be: ``For any given simplex, select a subset $\{ (x^{(k_0)}, y^{(k_0)}), \cdots, (x^{(k_{m'-1})}, y^{(k_{m'-1})}) \}$ from the training set $\{ (x^{(0)}, y^{(0)}), \cdots, (x^{(m-1)}, y^{(m-1)}) \}$ where $m'$ is the cardinality of subset, $m$ is the cardinality of subset, and all subset elements reside within the given simplex."
>
> >**Other suggestions: a） In Eqn(2), the ‘i’ is suggested to be replaced with 'n'; b） ”given function F(x) to be fitted” ->``given function F(x) to be fitted” c） The definition of SVR is not given.**
>
> We want to express our gratitude for the reviewer's valuable suggestions. We have incorporated these recommendations into the content of the newly revised paper.

---

> ### Author Response · Authors · 2023-11-20
> **Reply to Reviewer P3jk (References)**
>
> [1] Hashash, Y. M. A., Jung, S., & Ghaboussi, J. (2004). Numerical implementation of a neural network based material model in finite element analysis. International Journal for numerical methods in engineering, 59(7), 989-1005.
>
> [2] Javadi, A. A., Tan, T. P., & Zhang, M. (2003). Neural network for constitutive modelling in finite element analysis. Computer Assisted Mechanics and Engineering Sciences, 10(4), 523-530.

---

### Author Response · Authors · 2023-11-20
**General Response**

We would like to express our sincere gratitude to all the reviewers for their valuable feedback and for recognizing the strengths of our work. We appreciate their insightful comments, which have significantly contributed to enhancing the quality of our paper. Specifically, we are pleased that the reviewers have acknowledged the following key aspects:

1. **Novel Perspective:** Reviewers P3jk and YmUG have noted the novel perspective we bring to unsupervised representation learning.
2. **Learning-Free Method:** Reviewers RtZ7 and YmUG have appreciated that our encoder does not require any training.
3. **Theoretical Analyses:** Reviewers P3jk, RtZ7, and YmUG have recognized the value of the detailed theoretical analyses provided in our work.
4. **Generality:** Reviewer RtZ7 has highlighted the generality of our method, which can be evaluated across various tasks, from traditional data fitting to computer vision and NLP tasks.

In response to the reviewers' constructive feedback, we have made significant improvements to our manuscript and supplementary materials, including the following key changes:

1. **Extensive Experiments:** We have included extensive experiments on diverse datasets, as Appendix E outlines.
2. **Ablation Studies:** We have conducted additional ablation studies in Appendix E, providing experimental evidence that our train-free LagrangeEmbedding encoder can extract more complex features.
3. **Cost-Efficiency Solution:** We have proposed and evaluated a cost-efficient solution to accelerate the LagrangeEmbedding encoder in Appendix E.
4. **Comparison with Kernel Methods:** We have added a comparison of the LagrangeEmbedding-based network with kernel methods in Appendix F.
5. **Modular and Parallel Design:** We have redesigned the LagrangeEmbedding encoder to be modular and parallel, allowing it to be used independently or seamlessly integrated into any architecture as a plug-and-play module. The upgraded encoder enhances feature extraction and overall model performance.

These changes address the reviewers' comments and further strengthen the contributions and applicability of our work. We believe these revisions have significantly improved the quality and impact of our paper.

---

### Meta-Review · Area_Chair_mtSv · 2023-12-09

**Metareview:**

The submission proposes a piecewise linear function computed directly from training data as a competitor to neural networks.  The submission received borderline reviews with two feeling it could be slightly above the threshold and one feeling that it fell below the threshold of acceptance.  The premise of the approach is that a Lagrange embedding be used to give a piecewise linear function based purely on training data (in this sense it feels quite like a non-parametric method, and the term "untrainable" as used by the authors in the abstract is not necessarily very informative).  It is also well known that nearest neighbor methods work quite well.  Two main related concerns remain after the author-reviewer discussion: (i) how does this method scale to large scale data, and (ii) is it computationally competitive with NNs also at inference time.  In response to reviewer concerns about comptuation the authors only provided "init time", which is not representative of the full inference cost for applying the method.  Big-O analysis of computational complexity is not provided.

**Justification For Why Not Higher Score:**

The idea is interesting, but scalability is far from demonstrated.  Additional details of computational complexity and demonstration on a larger scale real-world task would be necessary.

**Justification For Why Not Lower Score:**

N/A

---

### Decision · Program_Chairs · 2024-01-16

Reject